# Lotus (*Nelumbo nucifera* Gaertn.) and Its Bioactive Phytocompounds: A Tribute to Cancer Prevention and Intervention

**DOI:** 10.3390/cancers14030529

**Published:** 2022-01-21

**Authors:** Anupam Bishayee, Palak A. Patel, Priya Sharma, Shivani Thoutireddy, Niranjan Das

**Affiliations:** 1College of Osteopathic Medicine, Lake Erie College of Osteopathic Medicine, Bradenton, FL 34211, USA; ppatel24886@med.lecom.edu (P.A.P.); PSharma44656@med.lecom.edu (P.S.); SThoutired89922@med.lecom.edu (S.T.); 2Department of Chemistry, Iswar Chandra Vidyasagar College, Belonia 799155, Tripura, India; ndnsmu@gmail.com

**Keywords:** *Nelumbo nucifera*, phytochemicals, cancer, prevention, therapeutic benefits, molecular mechanisms

## Abstract

**Simple Summary:**

The plant *Nelumbo nucifera* (Gaertn.), commonly known as lotus, sacred lotus, Indian lotus, water lily, or Chinese water lily, is an aquatic perennial crop belonging to the family of Nelumbonaceae. *N. nucifera* has traditionally been used as an herbal medicine and functional food in many parts of Asia. It has been found that different parts of this plant consist of various bioactive phytocompounds. Within the past few decades, *N. nucifera* and its phytochemicals have been subjected to intense cancer research. In this review, we critically evaluate the potential of *N. nucifera* phytoconstituents in cancer prevention and therapy with related mechanisms of action.

**Abstract:**

Cancer is one of the major leading causes of death worldwide. Accumulating evidence suggests a strong relationship between specific dietary habits and cancer development. In recent years, a food-based approach for cancer prevention and intervention has been gaining tremendous attention. Among diverse dietary and medicinal plants, lotus (*Nelumbo nucifera* Gaertn., family Nymphaeaceae), also known as Indian lotus, sacred lotus or Chinese water lily, has the ability to effectively combat this disease. Various parts of *N. nucifera* have been utilized as a vegetable as well as an herbal medicine for more than 2000 years in the Asian continent. The rhizome and seeds of *N. nucifera* represent the main edible parts. Different parts of *N. nucifera* have been traditionally used to manage different disorders, such as fever, inflammation, insomnia, nervous disorders, epilepsy, hypertension, cardiovascular diseases, obesity, and hyperlipidemia. It is believed that numerous bioactive components, including alkaloids, polyphenols, terpenoids, steroids, and glycosides, are responsible for its various biological and pharmacological activities, such as antioxidant, anti-inflammatory, immune-modulatory, antiviral, hepatoprotective, cardioprotective, and hypoglycemic activities. Nevertheless, there is no comprehensive review with an exclusive focus on the anticancer attributes of diverse phytochemicals from different parts of *N. nucifera*. In this review, we have analyzed the effects of *N. nucifera* extracts, fractions and pure compounds on various organ-specific cancer cells and tumor models to understand the cancer-preventive and therapeutic potential and underlying cellular and molecular mechanisms of action of this interesting medicinal and dietary plant. In addition, the bioavailability, pharmacokinetics, and possible toxicity of *N. nucifera*-derived phytochemicals, as well as current limitations, challenges and future research directions, are also presented.

## 1. Introduction

Cancer is the second leading cause of both morbidity and mortality throughout the world, with an estimated 19.3 million new cancer cases and almost 10 million cancer deaths in the year 2020 [1]. Interestingly, it has been suggested that more than 40% of all cancer deaths could be prevented via lifestyle changes, including diet [2]. High dietary consumption of fruits and vegetables (more than 400 g/day) may prevent at least 20% of all cancers [3]. Dietary intervention may also improve the efficacy of cancer chemotherapy and lower the risk of long-term complications in cancer patients [4]. The cancer-preventive potential of various fruits, vegetables, spices, whole grains, and herbs is attributed to the presence of secondary plant metabolites, also known as phytochemicals. These naturally-occurring phytochemicals are also utilized in the discovery and development of anticancer drugs [5,6]. Emerging preclinical and clinical data shows that bioactive food components contain enormous cancer-preventive and anticancer therapeutic potential due to their unique ability to impact various cancer hallmarks, namely sustained proliferation, cell death resistance, energy metabolism, immune surveillance evasion, inflammation, invasion, angiogenesis, and metastasis, by modulating a plethora of oncogenic and oncosuppressive signaling pathways [7,8,9,10,11,12,13,14,15,16,17].

*Nelumbo nucifera* Gaertn., commonly known as lotus, sacred lotus, Indian lotus or Chinese water lily, is a well-known dietary and medicinal plant. It is a large, perennial aquatic plant that belongs to the family *Nelumbonaceae* and consists of a sole genus Nelumbo with two species, *N. nucifera* and *N. lutea*, which are called Asian lotus and American lotus, respectively [18,19]. In general, lotus refers to the Asian lotus, which is mainly distributed in India, China, Nepal, Sri Lanka, Thailand, Japan, New Guinea and Australia [20,21,22]. In contrast, the American lotus is predominantly found in the eastern and southern regions of North America and north of South America [18,23,24]. Since ancient times, the lotus flower (Figure 1) has been recognized as a spiritual object for Hindus, Buddhists and Egyptians, and is considered a symbol of longevity in Chinese traditional culture [25,26]. Lotus is the national flower of India and Vietnam. As a vegetable as well as a medicinal and ornamental plant, lotus has been cultivated for more than the last 7000 years [19]. Almost all parts of the lotus plant have been used as food and medicine for more than 2000 years in Asia [27]. Lotus has a significant economic value in various Asian countries, where it is emerging as a horticultural model plant [26]. China is the largest producer and consumer of lotus in the world [27]. The rhizome (modified stem), perianth (non-reproductive part of the flower) and seeds of lotus are popular food ingredients and are used in various food products due to their delicious taste and nutritional value [28,29].

Various parts of the lotus plant (Figure 1), from root to shoot, have documented use in different traditional systems of medicines, such as Indian traditional medicine (Ayurveda) and Chinese traditional medicine [30,31,32]. The whole plant, as well as crude extracts, fractions and constituents, have been found to possess numerous biological and pharmacological activities, including antioxidant, anti-inflammatory, immunomodulatory, antipyretic, antibacterial, antiviral, antifungal, antidiarrheal, diuretic, antiamnestic, antithrombotic, antiarrythmic, antidiabetic, hypocholesterolemic, antiobesity, antiaging, antiatherosclerotic, antifibrotic, sedative, antineurodegenerative, memory-improving, antifertility, hepatoprotective, skin-protective, cardiovascular-protective, and anticancer properties [24,25,29,31,33,34,35,36]. The remarkable health-promoting and disease-mitigating activities of the lotus plant have been correlated with the presence of numerous bioactive phytocompounds, including polyphenols, flavonoids, phenolic acids, alkaloids, terpenoids, steroids, fatty acids, and glycosides [25,31,33,34,35,36].

During the last several decades, *N. nucifera* has been subjected to intense research that evaluated the antineoplastic effects of various parts and active constituents of this dietary and medicinal plant. However, to the best of our knowledge, a comprehensive state-of-the-art review of all available anticancer studies has not been performed. Most of the previous reviews mainly focused on traditional and ethnopharmacological uses, biosynthesis, phytochemical analysis, industrial applications, and broad-spectrum health benefits of *N. nucifera* in which an overview of anticancer potential represents a minor representation [29,31,32,33,34,37]. Several prior publications highlighted pharmacological activities of selected phytochemicals of *N. nucifera,* and these reports did not have a sole focus on cancer [25,35,36,38,39]. Hence, the aim of this work has been to perform a systematic and critical analysis of fragmentary studies to provide an up-to-date and complete assessment of cancer-preventive and anticancer therapeutic attributes of *N. nucifera* and its bioactive phytocomponents with an understanding of the cellular and molecular mechanisms of action. Moreover, the bioavailability, pharmacokinetics and possible adverse effects of *N. nucifera*-derived phytochemicals, as well as current limitations, challenges and future research directions, are also discussed. 

## 2. Chemical Constituents of *N. nucifera*

Several parts of *N. nucifera* are known to contain various pharmacologically active constituents and the most prominent phytochemical classes include alkaloids, flavonoids, terpenoids polysaccharides, steroids, essential oils, tannins, glycosides, proteins, fatty acids, minerals, and vitamins [29,31,34,40,41,42,43]. The main bioactive constituents of *N. nucifera* are alkaloids and flavonoids [33]. Different parts of *N. nucifera* contain several types of phytochemicals. For example, the leaves are rich in flavonoids and alkaloids, the flowers and plumules are rich in flavonoids, the seeds are rich in alkaloids, and the rhizome is rich in starch [29,40,42,44]. Procyanidins are the chief active components of *N. nucifera* receptacles [29]. A list of diverse groups of major phytochemicals present in different parts of *N. nucifera* is provided in Table 1.

Various bioactive alkaloids in different parts of *N. nucifera* include *N*-nornuciferine, nuciferine, roemerin, 2-hydroxy-1-methoxyaporphine, (6*R*,6a*R*)-roemerine-*N*_β_-oxide, (*S*)-armepavine, (+)-1(*R*)-coclaurine, (−)-1(*S*)-norcoclaurine, lotusine, isoliensinine, liensinine, neferine, liriodenine, asimilobine, pronuciferine, oleracein E, demethyl-coclaurine, dauricine, *cis*-*N*-coumaroyltyramine, *cis*-*N*-feruloyltyramine, *trans*-*N*-coumaroyltyramine, and *trans*-*N*-feruloyltyramine (Figure 2) [31,33,34,40,44,45,46,47]. 

The different parts of the *N. nucifera* plant, such as leaves, roots, seeds, and flowers, contain several bioactive flavonoid molecules, including flavonols, flavons, flavan-3-ols, flavanons, and anthocyanins [36]. The flavonols found in *N. nucifera* are myricetin, quercetin, kaempferol, and isorhamnetin, whereas the flavon molecules are diosmetin, syringetin, apigenin, luteolin, and chrysoeriol [36]. The flavan-3-ol molecules include catechin, epicatechin, catechin rhamnoside, gallocatechin gallate, and epigallocatechin gallate, along with its dimer and polymers, such as procyanidin dimer B1 and elephantorrhizol. The flavanones include naringenin and taxifolin, and the anthocyanins are cyanidin, delphinidin, malvidin, petunidin, and peonidin [36]. The major bioactive flavonoids derived from different parts of *N. nucifera* are quercetin, kaempferol, apigenin, isorhamnetin, luteolin, myricetin, syringetin, and diosmetin (Figure 3) [33,36]. A total of 16 flavonoid *C*-glycosides and 56 flavonoid *O*-glycosides have been isolated from different parts of *N. nucifera* [36]. The main flavonoid *O*-glycosides in the different parts of *N. nucifera* include myrycetin 3-*O*-galactoside, myrycetin 3-*O*-glucoside, quercetin 3-*O*-arabinopyranosyl-(1→2)-galactopyranoside, myrycetin 3-*O*-glucuronide, quercetin 3-*O*-rhamnopyranosyl-(1→6)-glucopyranoside (rutine), quercetin 3-*O*-galactoside (hyperoside), quercetin 3-*O*-glucoside (isoquercitrin), kaempferol 3-*O*-robinobioside, quercetin 3-*O*-glucuronide, kaempferol 3-*O*-galactoside, isorhamnetin 3-*O*-rutinoside, kaempferol 3-*O*-glucoside (astragalin), syringetin 3-*O*-glucoside, isorhamnetin 3-*O*-glucoside, kaempferol 3-*O*-glucuronide, kaempferol 7-*O*-glucoside, diosmetin 7-*O*-hexose, and isorhamnetin 3-*O*-glucuronide [25,42,48]. The key flavonoid *C*-glycosides in the different parts of *N. nucifera* are luteolin 8-*C*-*β*-d-glucopyranoside (orientin), luteolin 6-*C*-*β*-d-glucopyranoside (isoorientin), apigenin 8-*C*-*β*-d-glucopyranoside (vitexin), apigenin 6-*C*-*β*-d-glucopyranoside (isovitexin), apigenin 6-*C*-*β*-d-glucopyranosyl-8-*C*-*β*-d-glucopyranoside, luteolin 6-*C*-*β*-d-glucopyranosyl-8-*C*-*β*-d-pentoside, luteolin 6-*C*-*β*-d-pentosyl-8-*C*-*β*-d-glucopyranoside, apigenin 6-*C*-*β*-d-glucopyranosyl-8-*C*-*β*-d-xylopyranoside, apigenin 6-*C*-*β*-d-xylopyranosyl-8-*C*-*β*-d-glucopyranoside, apigenin 6-*C*-*β*-d-glucopyranosyl-8-*C*-*β*-d-arabionoside, apigenin 6-*C*-*β*-d-arabionosyl-8-*C*-*β*-d-glucopyranoside, apigenin 6-*C*-*β*-d-glucopyranosyl-8-*C*-*β*-d-rhamnoside, and apigenin 6-*C*-*β*-d-rhamnosyl-8-*C*-*β*-d-glucopyranoside [25]. The *N. nucifera* leaves, roots, and seed kernels contain several phenolic acids that can be classified into two categories: hydroxybenzoic acids and hydroxycinnamic acids [36]. The phenolic acids under the group of hydroxycinnamic acids include caffeic acid, protocetechuic acid, chlorogenic acid, cinnamic acid, ferulic acid, *p*-coumaric acid, and sinapic acid. The hydroxybenzoic acids include gallic acid, hydroxybenzoic acid, *p*-hydroxybenzoic acid, syringic acid, and vanillic acid [36]. 

### 2.1. Leaves

The leaves of *N. nucifera* are rich in alkaloids and flavonoids [29,40,44,45,47]. The phytochemical analysis of EtOAc-soluble fraction of 80% MeOH extract of leaves of *N. nucifera* revealed the presence of 33 significant bioactive constituents, including 13 megastigmanes, 1 sesquiterpene, 8 alkaloids, and 11 flavonoids [45]. The various bioactive alkaloids in *N. nucifera* have raised significant interest due to their versatile chemical and biological activities [46]. The leaves of *N. nucifera* are a rich source of various bioactive alkaloids [44,45,46,47]. The 95% EtOH extract of leaves of *N. nucifera* yielded six potent bioactive alkaloids, namely (+)-1(*R*)-coclaurine, (−)-1(*S*)-norcoclaurine, nuciferine, liensinine, neferine, and isoliensinine [40]. The high-performance liquid chromatography (HPLC) coupled with ion trap/time-of-flight mass spectrometry (LC/MS-ITTOF) analysis of *N. nucifera* leaf extract reported the existence of five isoquinoline alkaloids: dehydronuciferine, *N*-nornuciferine, *O*-nornuciferine, nuciferine, and roemerine [49]. The non-aqueous capillary electrophoresis coupled with ultraviolet and mass spectroscopy (NACE-UV-MS) analysis is a method developed by Do et al. [50] to determine the major alkaloids, such as (−)-caaverine, (+)-isoliensinine, (+)-norarmepavine, (−)-armepavine, (−)-nuciferine, (−)-nornuciferine, and (+)-pronuciferine, present in *N. nucifera* leaves. One alkaloid, *N*-methylasimilobine, has also been reported from the MeOH extract of *N. nucifera* leaves [51].

The potential bioactive flavonoids in the leaves of *N. nucifera* are quercetin, kaempferol, and luteolin [45,51]. The methanol extracts of the leaves of *N. nucifera* revealed the presence of several flavonoid compounds, including catechin, quercetin, quercetin-3-*O*-glucopyranoside, quercetin-3-*O*-glucuronide, quercetin-3-*O*-galactopyranoside, kaempferol-3-*O*-glucopyranoside, and myricetin-3-*O*-glucopyranoside [52]. The bioactive quercetin-based flavonoids and glycosides in *N. nucifera* leaves are (+)-catechin, hyperoside, isoquercitrin, and astragalin [29,40]. The in vitro lipolysis assay of the leaf extract of *N. nucifera* identified the presence of five bioactive flavonoid molecules, namely quercetin 3-*O*-*α*-arabinopyranosyl-(1→2)-*β*-galactopyranoside, (+)-catechin, hyperoside, isoquercitrin, and astragalin [53]. 

The gas chromatography-mass spectrometry (GC-MS) analysis of hexane extract of the leaves of *N. nucifera* revealed the presence of 38 compounds, in which 15 compounds are the essential oil composition with 9,12,15-octadecatrienoic acid, linoleic acid ethyl ester, *n*-hexadecanoic acid, hexadecanoic acid, ethyl ester, and methyl (*Z*)-5,11,14,17-eicosatetraenoate [54]. The GC-MS analysis of essential oils of *N. nucifera* leaves detected the presence of 95 constituents, in which the major constituents are *l*-(+)-ascorbic acid 2,6-dihexadecanoate, *trans*-phytol, hexahydrofarnesyl acetone, pentadecyl acrylate, *β*-ionone, geranyl acetone, propionic acid decyl ester, farnesyl acetone, and heneicosane [41]. 

The analysis of a MeOH extract of the leaves of *N. nucifera* showed the presence of five norsesquiterpenes/megastigmanes, such as (*E*)-3-hydroxymegastigm-7-en-9-one, (3*S*,5*R*,6*S*,7*E*)-megastigma-7-ene-3,5,6,9-tetrol, dendranthemoside B, icariside B2, and sedumoside F1 [51]. Two bioactive triterpenes, namely alphitolic acid and maslinic acid, have also been extracted from the MeOH extract of *N. nucifera* leaves [51]. 

The leaves of *N. nucifera* also contain several nonvolatile organic acids, such as alphitolic, maslinic, gallic, tartaric, malic, and anisic acids [29,51,55]. 

### 2.2. Plumules

The plumule of *N. nucifera* contains 7.8% moisture, 4.2% ash, 12.5% crude oil and 26.3% protein [56]. The *N. nucifera* plumule oil is a rich source of several fatty acids. The major fatty acids in *N. nucifera* plumule oil are linoleic acid (50.4%) and palmitic acid (18.0%), followed by oleic acid (13.5%), behenic acid (6.8%), arachidic acid (3.30%), linolenic acid (3.0%), and stearic acid (2.90%) [56]. The major triglyceride components in *N. nucifera* plumule oil include linoleic acid-linoleic acid (12.80%), *β*-palmitic acid-linoleic acid (11.27%), *β*-oleic acid-linoleic acid (9.43%), *β*-palmitic acid-linoleic acid-oleic acid (8.58%), *β*-behenic acid-linoleic acid (8.32%), *β*-palmitic acid-oleic acid-linoleic acid (8.28%), *β*-palmitic acid-linoleic acid-arachidic acid (7.99%), *β*-palmitic acid-linoleic acid-palmitic acid (7.71%), and *β*-linoleic acid-oleic acid-linoleic acid (7.13%) [56]. 

The unsaponifiable components in *N. nucifera* plumule oil are very high (up to 14–19%) due to the presence of several sterol compounds [56]. The major sterols present in *N. nucifera* plumule oil are *β*-sitosterol (31.75%), ∆5-avenasterol (19.66%), and campesterol (6.28%), followed by stigmasta-7,25-dien-3-ol (3.49%), 24-methylene-9,19-cyclolanostan-3*β*-ol (3.79%), *α*-sitosterol (3.41%), stigmasterol (2.67%) ergosta-8,24(28)-dien-3-ol (2.5%), ergosta-5,24-dien-3-ol (1.71%), lanost-7-en-3-one (1.51%), and stigmasta-7-en-3-ol (0.96%) [56]. 

The ultra-performance liquid chromatography coupled with quadrupole time of flight mass spectrometry analysis of *N. nucifera* plumule extracts revealed the presence of several bioactive constituents which are mainly of the alkaloid and flavonoid group of compounds [57]. The bioactive alkaloid components in *N. nucifera* plumule extracts include higenamine, lotusine, 4′-methylcoclaurine, isoliensinine, liensinine, neferine, and nuciferine [57]. Duan and Jiang [58] reported a new benzylisoquinoline alkaloid, namely nelumstemine [1-(4′-hydroxybenzoyl)-6,7-dimethoxy-3,4-dihydroisoquinoline], which was isolated from the stems of *N. nucifera*. The bioactive flavonoid constituents in *N. nucifera* plumule extracts include apigenin-6-*C*-*α*-l-glucopyanosyl-8-*C*-*β*-d-glucopyranoside and apigenin-6-*C*-*α*-l-arabofuranosyl-8-*C*-*β*-d-glucopyranoside [57]. 

### 2.3. Seeds and Rhizomes

Starch is the major ingredient (up to 70% of the dry weight) of both the rhizome and seeds of *N. nucifera* [28]. The raw and dried rhizome and seeds of *N. nucifera* are good sources of niacin and vitamin B6 [28]. The raw and dried portions of the rhizome and seeds of *N. nucifera* contain various minerals, vitamins and fatty acids [28]. The major minerals present in the seeds and dried rhizome of *N. nucifera* include potassium (16,300 µg/g), phosphorus (1715 µg/g), and magnesium (1650 µg/g), followed by aluminum (470 µg/g), calcium (445 µg/g), manganese (57 µg/g), sodium (33 µg/g), cobalt (16 µg/g), strontium (15), iron (13 µg/g), zinc (13 µg/g), copper (10 µg/g) and vanadium (7 µg/g) [24,28]. Other important nutritional agents are fat (72.17%), crude fiber (10.60%), moisture (10.50%), total ash (4.5%), proteins (2.7%), and crude carbohydrates (1.93%) [59]. The seed possesses higher energy with 348.45 cal per 100 g [59]. Vitamin C and folate are the main vitamins present in both the raw rhizome and seeds of *N. nucifera* [28]. *N. nucifera* seeds also contain several vitamins in large amounts, such as vitamin B1 (2.24 mg/kg), vitamin B2 (0.13 mg/kg), vitamin B6 (3.03 mg/kg), vitamin C (39.4 mg/kg), and vitamin E (4.6 mg/kg) [24]. 

Alkaloids are the key secondary metabolites in the seeds of *N. nucifera* [31]. The major alkaloids present in the seeds of *N. nucifera* are lotusine, isoliensinine, liensinine, dauricine, pronuciferine, nuciferine, procyanidin, neferine, roemerine, and armepavine [31]. The chloroform fraction of hot MeOH extract of the embryos of seeds of *N. nucifera* led to the isolation of three novel bisbenzylisoquinoline alkaloids, such as nelumboferine, nelumborines A and B, along with four previously reported alkaloids [55]. 

The protein, minerals, amino acids, and unsaturated fatty acids are the rich sources of the seeds of *N. nucifera* [31]. The seeds of *N. nucifera* also contain mainly four types of monosaccharide, namely D-galactose, L-arabinose, D-mannose and D-glucose [59]. The main nutrients in *N. nucifera* seeds are proteins and carbohydrates [24]. The level of polysaccharides in the rhizome of *N. nucifera* is very high [29]. Two antioxidant micromolecular constituents, such as (±)-gallocatechin and (−)-catechin, along with antioxidant macromolecular constituents as well as a polysaccharide–protein complex having *α*/*β*-pyranose and *α*-furanose ring of molecular mass 18.8 kDa, have been reported from the rhizome of *N. nucifera* [60]. The polysaccharide–protein complex is composed of mannose, rhamnose, glucose, galactose and xylose in the molar ratio of 2:8:7:8:1 [61]. The abundance of essential amino acids in *N. nucifera* seeds is very high [24]. The major essential amino acids present in the seeds of *N. nucifera* are methionine (1.64%), valine (1.10%), isoleucine (0.99%), lysine (0.97%), phenylalanine (0.86%), histidine (0.50%), threonine (0.45%), and leucine (0.15%) [24]). The key carbohydrates of *N. nucifera* seeds are starch, polysaccharides and oligosaccharides [24]. The main nutrients of *N. nucifera* seeds represent starch (55.77%), crude protein (16.2%), water (14.0%), sugar (8.13%), ash (4.05%) and fat (2.05%) with a total calorific value of 1432 kJ/100 g [24]. 

*N. nucifera* seeds are also rich sources of various fatty acids, such as 14-methylpentadecanoic acid, 8,11-octadecadienoic acid, anti-9-octadecenoic acid, 18-carbonate, behenic acid, 20-carbonate, 9,12,15-octadecatrienoic acid, 14-carbonate, 23-carbonate, pentadecanoate, 17-carbonate, maleic-7-hexadecene acid, anti-8-octadecenoic, and maleic-9-octadecenoic acid [24]. 

From the MeOH extract of *N. nucifera* rhizome, a new ursane triterpenoid ester, urs-12-*en*-3*β*-*O*-9*E*,12*E*-octadecadienoate, was isolated [62]. The fresh seed epicarps of *N. nucifera* are rich in flavonols and their glycosides. The most important flavonols in *N. nucifera* epicarps are myricetin, quercetin, kaempferol, and isorhamnetin [63]. The CC along with RP-HPLC and HPLC–ESI-MS analysis of *N. nucifera* seed epicarp at three different ripening stages, the green ripening stage, half ripening stage, and full ripening stage, revealed the presence of four bioactive polyphenolic compounds: catechin, epicatechin, hyperoside, and isoquercitrin [64]. It is reported that the levels of these four bioactive polyphenolic compounds are different during seed ripening. The levels of catechin and epicatechin decrease in the ripening stage, whereas the levels of hyperoside and isoquercitrin increase during the ripening stage [64]. The rhizome of *N. nucifera* contains considerable amounts of various phytochemicals, such as tannins, saponins, and phenolic acids [65]. The major phytochemicals present in *N. nucifera* embryo include liensinine, isoliensinine, neferine, nuciferine, lotusine, pronuciferine, rutin, hyperin, and demethylcoclaurine [33].

### 2.4. Flowers

Several classes of bioactive phytochemicals, such as polyphenols, flavonoids, tannins and terpenoids, have been isolated from the ethanolic extracts of *N. nucifera* pink flower stamens and petals [43]. Nakamura et al. [66] reported the presence of a new alkaloid, namely *N*-methylasimilobine *N*-oxide, along with 11 bioactive benzylisoquinoline alkaloids, in the methanolic extracts of flower buds and leaves of *N. nucifera*. The major phytochemicals present in the flowers of *N. nucifera* are quercetin, luteolin, luteolin glucoside, kaempferol, kaempferol-3-*O*-glucoside, and isoquercitrin [33]. The ethanolic extract of *N. nucifera* petals revealed the presence of nine potent bioactive benzylisoquinoline alkaloids: (+)-juziphine, (+)-isococlaurine, (−)-*N*-methylisococlaurine, (−)-*N*-methylcoclaurine, (+)-nor-roefractine, (+)-armepavine, (−)-caaverine, (−)-lirinidine, and (+)-glaziovine [67]. The colors of the petals of *N. nucifera* are different only due to the presence of two flavonoids, namely isorhamnetin and kaempferol [68].

**Table 1 cancers-14-00529-t001:** List of phytochemicals reported from the different parts of *N. nucifera*.

Phytochemicals	Plant Part	References
*Aromatic phenolic compounds*		
Arbutin	Stamens	Mukherjee et al., 2009 [31]
Gallic acid	Stamens and petals	Noysang and Boonmatit, 2019 [43]
(*E*)-Ferulic acid	Seeds	Rho and Yoon, 2017 [69]
(*E*)-*p*-Coumaric acid	Seeds	Rho and Yoon, 2017 [69]
(*E*)-Sinapate-4-*O*-*β*-d-glucopyranoside	Seeds	Rho and Yoon, 2017 [69]
*p*-Hydroxybenzoic acid	Seeds	Rho and Yoon, 2017 [69]
Protocatechuic acid	Seeds	Rho and Yoon, 2017 [69]
Tannic acid	Stamens and petals	Noysang and Boonmatit, 2019 [43]
*Megastigmane/sesquiterpenes compounds*		
(−)-Boscialin	Leaves	Ahn et al., 2013 [45]
(+)-Dehydrovomifoliol	Leaves	Ahn et al., 2013 [45]
(+)-Epiloliolide	Leaves	Ahn et al., 2013 [45]
(*E*)-3-Hydroxymegastigm-7-en-9-one	Leaves	Ahn et al., 2013 [45]
3-oxo-Retro-*α*-ionol I	Leaves	Ahn et al., 2013 [45]
3*S*,5*R*-Dihydroxy-6*S*,7-megastigmadien-9-one	Leaves	Ahn et al., 2013 [45]
5,6-epoxy-3-Hydroxy-7-megastigmen-9-one	Leaves	Ahn et al., 2013 [45]
Annuionone D	Leaves	Ahn et al., 2013 [45]
Byzantionoside A	Leaves	Ahn et al., 2013 [45]
Grasshopper ketone	Leaves	Ahn et al., 2013 [45]
Icariside B2	Leaves	Ahn et al., 2013 [45]
Nelumnucifoside A	Leaves	Ahn et al., 2013 [45]
Vomifoliol	Leaves	Ahn et al., 2013 [45]
Nelumnucifoside B	Leaves	Ahn et al., 2013 [45]
*Alkaloids*		
(−)-1(*R*)-*N*-methylcoclaurine	Leaves	Kashiwada et al., 2005 [40]
(−)-Lirinidine (5-demethylnuciferine)	Flower buds, stamen, and leaves	Nakamura et al., 2013 [33]; Paudel & Panth, 2015 [33]
(−)-Anonaine	Leaves	Wang et al., 2011 [70]
(−)-Asimilobine	Leaves	Wang et al., 2011 [70]
(−)-Caaverine	Leaves	Wang et al., 2011 [70]
(−)-*N*-Methylasimilobine	Leaves	Wang et al., 2011 [70]
(−)-*nor*-Nuciferine	Leaves	Wang et al., 2011 [70]
(−)-Nuciferine	Leaves	Wang et al., 2011 [70]
(−)-Roemerine	Leaves	Wang et al., 2011 [70]
(6*R*,6a*R*)-Roemerine-*N*_β_-oxide	Leaves	Ahn et al., 2013 [45]
(*R*)-Roemerine	Leaves	Agnihotri et al., 2008 [71]
2-Hydroxy-1-methoxy-6a,7-dehydroaporphine	Flower buds and leaves	Nakamura et al., 2013 [66]
3-Indoleacetic acid	Seeds	Rho and Yoon, 2017 [69]
4′-Methyl-*N*-methylcoclaurine	Plumule	Zhou et al., 2013 [57]
7-Hydroxydehydronuciferine	Leaves	Wang et al., 2011 [69]
Anisic acid	Seeds	Itoh et al., 2011 [55]
Anonaine	Leaves	Agnihotri et al., 2008 [71]
Anonaine	Stamen	Paudel & Panth, 2015 [33]
Armepavine	Plumule	Zhou et al., 2013 [57]
Armepavine	Stamen	Paudel & Panth, 2015 [33]
Asimilobine	Flower buds and leaves	Nakamura et al., 2013 [66]
Asimilobine	Stamen	Paudel & Panth, 2015 [33]
Cepharadione B	Leaves	Wang et al., 2011 [70]
*cis*-*N*-Coumaroyltyramine	Leaves	Ahn et al., 2013 [45]
*cis*-*N*-Feruloyltyramine	Leaves	Ahn et al., 2013 [45]
Coclaurine	Seeds	Kashiwada et al., 2005 [40]
*d,l*-Armepavine	Flower buds and leaves	Kashiwada et al., 2005 [40]; Nakamura et al., 2013 [65]
Dauricine	Seeds	Paudel & Panth, 2015 [33]
Dehydroanonaine	Stamen	Paudel & Panth, 2015 [33]
Dehydroemerine	Stamen	Paudel & Panth, 2015 [33]
Dehydronuciferine	Flower buds and leaves	Nakamura et al., 2013 [65]
Dehydronuciferine	Stamen	Paudel & Panth, 2015 [33]
Dehydroroemerine	Leaves	Agnihotri et al., 2008 [71]
Demethylcoclaurine	Stamen	Paudel & Panth, 2015 [33]
Higenamine	Plumule	Zhou et al., 2013 [57]
Higenamine 4′-*O*-*β*-d-glucoside	Plumule	Kato et al., 2015 [72]
Isoliensinine	Seeds, leaves, and stamen	Itoh et al., 2011 [55]; Kashiwada et al., 2005 [40]; Paudel & Panth, 2015 [33]
Liensinine	Seeds, leaves, and stamen	Itoh et al., 2011 [55]; Kashiwada et al., 2005 [40]; Paudel & Panth, 2015 [33]
Liriodenine	Leaves and stamen	Ahn et al., 2013 [45]; Wang et al., 2011 [70]; Paudel & Panth, 2015 [33]
Lotusine	Leaves and seeds	Kashiwada et al., 2005; Paudel & Panth, 2015 [33]
Lysicamine	Flower buds, leaves, and leaves	Wang et al., 2011 [70]; Nakamura et al., 2013 [66]
Neferine	Seeds and leaves	Kashiwada et al., 2005 [40]; Itoh et al., 2011 [55]; Paudel & Panth, 2015 [33]
*N*-methylasimilobine	Leaves	Agnihotri et al., 2008 [71]
*N*-Methylasimilobine	Flower buds, stamen, and leaves	Nakamura et al., 2013 [66]; Paudel & Panth, 2015 [33]
*N*-Methylcoclaurine	Stamen	Paudel & Panth, 2015 [33]
*N*-Methylisococlaurine	Stamen	Paudel & Panth, 2015 [33]
*N*-Norarmepavine	Stamen	Paudel & Panth, 2015 [33]
*N*-Nornuciferine	Flower buds and leaves	Nakamura et al., 2013 [65]
Norjuziphine	Flower	Morikawa et al., 2016 [73]
Nornuciferine	Leaves and stamen	Kashiwada et al., 2005 [40]; Paudel & Panth, 2015 [33]
Nuciferine	Flower buds, leaves, seeds, and leaves	Kashiwada et al., 2005 [40]; Agnihotri et al., 2008 [71]; Nakamura et al., 2013 [66]; Paudel & Panth, 2015 [33]
Nuciferine *N*-oxide	Flower buds and leaves	Nakamura et al., 2013 [66]
Oleracein E	Leaves	Ahn et al., 2013 [45]
*O*-Nornuciferine	Plumule	Zhou et al., 2013 [57]
Pronuciferine	Leaves	Ahn et al., 2013 [45]; Nakamura et al., 2013 [66]
Reserpine	Stamens and petals	Noysang and Boonmatit, 2019 [43]
Roemerin	Stamen, leaves, and plumule	Kashiwada et al., 2005 [40]; Zhou et al., 2013 [57]; Paudel & Panth, 2015 [33]
*trans*-*N*-Coumaroyltyramine	Leaves	Ahn et al., 2013 [45]
*trans*-*N*-Feruloyltyramine	Leaves	Ahn et al., 2013 [45]
Tryptophan	Seeds	Rho and Yoon, 2017 [69]
*Flavonoids*		
(−)-Catechin	Leaves	Ahn et al., 2013 [45]
5,7,3′,5′-Tetrahydroxyflavanone	Leaves	Ahn et al., 2013 [45]
Chrysoeriol 7-*O*-*β*-d-glucopyranoside	Leaves	Wang et al., 2008 [74]; Ahn et al., 2013 [45]
Elephantorrhizol	Leaves	Ahn et al., 2013 [45]
Epitaxifolin	Leaves	Ahn et al., 2013 [45]
Hyperoside	Leaves, and plumule	Wang et al., 2008 [74]; Kashiwada et al., 2005 [40]; Zhou et al., 2013 [57]; Liu et al., 2016 [75]
Isoquercitrin (Hirsutrin)	Receptacles, Stamen, plumule, and leaves	Wang et al., 2008 [73]; Kashiwada et al., 2005 [40]; Zhou et al., 2013 [57]; Paudel & Panth, 2015 [33]; Liu et al., 2016 [75]
Isorhamnetin	Leaves	Wang et al., 2008 [74]
Isorhamnetin 3-*O*-*β*-d-glucopyranoside	Receptacles and stamens	Mukherjee et al., 2009 [31]; Liu et al., 2016 [75]
Isorhamnetin 3-*O*-rutinoside	Leaves	Kihyun et al., 2009 [51]
Isorhamnetin 3-*O*-*α*-l-rhamnopyranosyl-(1→6)-*β*-d-glucopyranoside	Stamens	Mukherjee et al., 2009 [31]
Isoschaftoside	Seeds	Rho and Yoon, 2017 [69]
Kaempferol	Leaves and stamens	Ahn et al., 2013 [45]; Wang et al., 2008 [74]; Mukherjee et al., 2009 [31]
Kaempferol 3-*O*-robinobioside	Receptacles	Liu et al., 2016 [75]
Kaempferol 3-*O*-*α*-l-rhamnopyranosyl-(1→2)-*β*-d-glucopyranoside	Stamens	Mukherjee et al., 2009 [31]
Kaempferol 3-*O*-*α*-l-rhamnopyranosyl-(1→2)-*β*-d-glucuronopyranoside	Stamens	Mukherjee et al., 2009 [31]
Kaempferol 3-*O*-*α*-l-rhamnopyranosyl-(1→6)-*β*-d-glucopyranoside	Stamens	Mukherjee et al., 2009 [31]
Kaempferol 3-*O*-*β*-d-galactopyranoside	Receptacles and stamens	Mukherjee et al., 2009 [31]; Liu et al., 2016 [75]
Kaempferol 3-*O*-*β*-d-glucopyranoside/astragalin	Receptacles, leaves, and stamens	Wang et al., 2008 [74]; Mukherjee et al., 2009 [31]; Ahn et al., 2013 [45]; Liu et al., 2016 [75]
Kaempferol 3-*O*-*β*-d-glucuronopyranoside	Stamens	Mukherjee et al., 2009 [31]; Paudel & Panth, 2015 [33]
Kaempferol 3-*O*-*β*-d-glucuronopyranosyl methylester	Stamens	Mukherjee et al., 2009 [31]
Kaempferol 7-*O*-*β*-d-glucopyranoside	Stamens	Mukherjee et al., 2009 [31]
Luteolin/luteolin glucoside	Leaves, plumule, and stamen	Ahn et al., 2013 [45]; Zhou et al., 2013 [57]; Paudel & Panth, 2015 [33]
Myricetin 3′,5′-dimethylether 3-*O*-*β*-d-glucopyranoside	Stamens	Mukherjee et al., 2009 [31]
Myricetin 3-*O*-galactoside	Receptacles	Liu et al., 2016 [75]
Myricetin 3-*O*-glucoside	Receptacles	Liu et al., 2016 [75]
Myricetin 3-*O*-glucuronide	Receptacles	Liu et al., 2016 [75]
Nelumboroside A	Stamens	Mukherjee et al., 2009 [31]
Nelumboroside B	Stamens	Mukherjee et al., 2009 [31]
Quercetin	Leaves, stamens and petals	Wang et al., 2008 [74]; Ahn et al., 2013 [45]; Paudel & Panth, 2015 [33]; Liu et al., 2016 [75]; Noysang and Boonmatit, 2019 [43]
Quercetin 3-*O*-glucuronide/Quercetin 3-*O*-*β*-d-glucuronide	Receptacles and leaves	Kashiwada et al., 2005 [40]; Kihyun et al., 2009 [51]; Liu et al., 2016 [75]
Quercetin 3-*O*-*β*-d-glucopyranoside	Leaves and stamens	Agnihotri et al., 2008 [71]; Mukherjee et al., 2009 [31]; Kihyun et al., 2009 [51]; Ahn et al., 2013 [45]
Quercetin 3-*O*-*β*-d-xylopyranosyl-(1→2)-*β*-d-galactopyranoside	Leaves	Kashiwada et al., 2005 [40]
quercetin-3-*O*-*β*-d-xylopyranosyl-(1→2)-*β*-d-glucopyranosyl glycoside	Leaves	Wang et al., 2008 [74]
Rutin	Leaves, stamens and petals	Kashiwada et al., 2005 [40]; Noysang and Boonmatit, 2019 [43]
Syringetin 3-*O*-glucoside	Receptacles	Liu et al., 2016 [75]
Taxifolin	Leaves	Ahn et al., 2013 [45]
*Sterols and triterpenoids*		
24(*R*)-Ethylcholest-6-ene-5*α*-ol-3-*O*-*β*-d-glucopyranoside	Leaves	Agnihotri et al., 2008 [71]
Stigmasta-4,22-dien-3-one	Leaves	Wang et al., 2011 [70]
*Β*-Sitostenone	Leaves	Wang et al., 2011 [70]
*β*-Sitosterol	Rhizome	Chaudhuri & Singh, 2009 [61]
*β*-Sitosterol-3-*O*-glucoside/*β*-Sitosterol-3-*O*-*β*-d-glucopyranoside	Rhizome, leaves, and stamens	Agnihotri et al., 2008 [71]; Chaudhuri & Singh, 2009 [62]; Mukherjee et al., 2009 [31]
Betulinic acid	Rhizome	Chaudhuri & Singh, 2009 [61]
*α*-Amyrin	Rhizome	Chaudhuri & Singh, 2009 [61]
*Aliphatic open chain compounds*		
10-Eicosanol	Leaves	Agnihotri et al., 2008 [71]
3,7,11,15-Tetramethyl-1-hexadecen- 3-ol (isophytol)	Leaves	Agnihotri et al., 2008 [71]
3,7,11,15- Tetramethyl-2-hexadecen-1-ol (*trans*-phytol)	Leaves	Agnihotri et al., 2008 [71]
7,11,15-Trimethyl-2-hexadecanone	Leaves	Agnihotri et al., 2008 [71]
Nonacosan-10-ol	Leaves	Mukherjee et al., 2009 [31]
Triacontan-7-ol	Leaves	Mukherjee et al., 2009 [31]
Nonacosane-4,10-diol	Leaves	Mukherjee et al., 2009 [31]
Nonacosane-5,10-diol	Leaves	Mukherjee et al., 2009 [31]
Nonacosane-10,13-diol	Leaves	Mukherjee et al., 2009 [31]
Hentriacontane-12,15-diol	Leaves	Mukherjee et al., 2009 [31]
Tritriacontane-9,10-diol	Leaves	Mukherjee et al., 2009 [31]
Octadecanoic acid	Leaves	Mukherjee et al., 2009 [31]
Palmitic acid	Rhizome	Chaudhuri & Singh, 2009 [62]
Linoleic acid	Rhizome	Chaudhuri & Singh, 2009 [62]
9*E*,12*E*,15*E*-Octadecatrienoic acid	Rhizome	Chaudhuri & Singh, 2009 [62]
Linalool	Stamen	Paudel & Panth, 2015 [33]
Tartaric acid	Leaves	Paudel & Panth, 2015 [33]
Gluconic acid	Leaves	Paudel & Panth, 2015 [33]
Acetic acid	Leaves	Paudel & Panth, 2015 [33]
Malic acid	Leaves	Paudel & Panth, 2015 [33]
Ginnol	Leaves	Paudel & Panth, 2015 [33]
Nonadecane	Leaves	Paudel & Panth, 2015 [33]
Succinic acid	Leaves	Paudel & Panth, 2015 [33]
*Miscellaneous compounds*		
Dihydrophaseic acid	Seeds	Rho and Yoon, 2017 [69]
Dihydrophaseic acid 3′-*O*-*β*-d-glucopyranoside	Seeds	Rho and Yoon, 2017 [69]
Pheophytin-a (chlorophyll derivative)	Leaves	Wang et al., 2011 [70]
Aristophyll-C (chlorophyll derivative)	Leaves	Wang et al., 2011 [70]

## 3. *N. nucifera* Extracts, Fractions and Pure Compounds in Cancer Research

### 3.1. Literature Search Methodology

The Preferred Reporting Items for Systematic Reviews and Meta-Analysis (PRISMA) criteria [76], which is recommended for reporting systematic reviews, was followed for this work. The literature search was conducted using various databases, such as PubMed, ScienceDirect, Google Scholar, and Scopus. Publication data was not a criterion when filtering research articles. The last search was performed in December 2021. Various combinations of keywords that were used included: *Nelumbo nucifera*; lotus; cancer; in vivo, in vitro; tumor; prevention; treatment, proliferation, apoptosis, and clinical studies. Reviews, systemic reviews, meta-analyses, letters to editors, book chapters, conference abstracts, and unpublished results were not included, and only articles written in the English language were considered. Additionally, available clinical studies pertaining to *N. nucifera* and cancer were searched on clinicaltrials.gov. The quality of each animal study was evaluated according to SYRCLE’s RnB tool that investigates sources of bias [77]. Reference lists from reviews and collected articles were checked for relevant publications. Articles not related to cancer or *N. nucifera* have been excluded. Upon applying these exclusion criteria, eligible full-length articles were obtained and reviewed, and a collaborative decision was made for their incorporation for further analysis. Figure 4 illustrates the overview of the literature search and study selection.

### 3.2. Preclinical Studies (In Vitro and In Vivo)

Potential anticancer effects and mechanisms of action of *N. nucifera*-derived extracts, fractions and pure compounds have been investigated using various cancer cell lines and animal tumor models. Table 2 and Table 3 summarize relevant in vitro and in vivo results, respectively.

#### 3.2.1. Breast Cancer

The anticancer properties of various parts of the *N. nucifera* plant have been extensively investigated using different breast cancer cell lines. Karki et al. [78] explored the anticancer properties of the aqueous rhizome extract of *N. nucifera* on the MDA-MB-231 human breast cancer cell line. It was found that the extract inhibited the proliferation and migration of those cells by decreasing the levels of matrix metalloproteinase-2 (MMP-2) and MMP-9 (Table 2). Yang et al. [79] treated MCF-7 human breast cancer cells with flavonoid-rich aqueous *N. nucifera* extract (NLE) and found that it inhibited the proliferation of MCF-7 cells by arresting them in the G0/G1 phase of the cell cycle. Mechanistically, NLE induced p53 phosphorylation (p-p53), increased levels of cyclin-dependent kinase inhibitory proteins, such as p21, p27, and p16, downregulated cyclin expressions, and decreased levels of cyclin D1/cyclin-dependent kinase 4 (CDK4) and cyclin E/CDK2 complexes which led to the upregulation of Rb/E2F pathway and ultimately resulted in a G1 phase arrest. Another important observation was the NLE-mediated inactivation of fatty acid synthase (Fas). Quantitative HPLC analysis revealed gallic acid as the most abundant flavonoid within the extracts, followed by rutin. In a follow-up study, the same research group [80] treated MDA-MB-231 human breast cancer cells with the aqueous leaf extract used in the previous study (NLE) to determine its anticancer effect. NLE treatment reduced cell viability, migratory and invasive properties by decreasing the levels of phosphorylated forms of extracellular signal-regulated kinase (p-ERK), phosphoinositide-3-kinase (p-PI3K), protein kinase B (p-Akt) as well as rat sarcoma gene (RAS) and mitogen-activated extracellular signal-regulated kinase (MEK). In addition, NLE inhibited angiogenesis by downregulating the expression of connective tissue growth factor (CTGF), MMP-2, vascular endothelial growth factor (VEGF) and nuclear factor-κB (NF-κB) p65, while increasing levels of tissue inhibitor of metalloproteinase-2 (TIMP-2). In an extension of previous studies, Wu et al. [81] observed that NLE and *N. nucifera* leaf polyphenol extract (NLPE) reduced cell viability and suppressed cell proliferation in 4T-1 breast cancer cells and exhibited anti-invasive and antimigratory effects in MDA-MB-231 and 4T-1 cells. These effects occurred mechanistically through increasing apoptosis, decreasing levels of Ras homolog family member A (RhoA), Ras-related C3 botulinum toxin substrate 1 (Rac1), cyclin-dependent kinase 42 (Cdc42) and suppressing activation of ERK1/2, p38 MAPK. Additionally, NLE and NLPE reduced the expression of protein kinase Cα (PKCα) in 4T-1 cells. Arjun et al. [82] used methanol and acetone leaf extracts of *N. nucifera* to investigate their anticancer effects on MCF-7 human breast cancer cells. Although both leaf extracts inhibited proliferation and reduced viability of human breast cancer cells, the methanol extract produced the maximum inhibitory effect. However, the underlying mechanism of action of these effects has not been reported. 

In addition to the leaf, other parts of the *N. nucifera* plant have also been studied for their anticancer activities. For example, *N. nucifera* flower receptacles have been investigated by Krubha and Vasan [83]. A methanolic extract from *N. nucifera* floral receptacles has shown antiproliferative and cytotoxic activities against MCF-7 human breast cancer cells by increasing antioxidant activity as demonstrated by 2,2-diphenyl-1-picrylhydrazyl radical scavenging assay.

Various researchers used pure compounds isolated from the *N. nucifera* plant to investigate their anticancer properties against breast cancer in vitro. In one study, MDA-MB-231 human breast cancer cells were treated with isoliensinine, liensinine and neferine. Although all three compounds exhibited antiproliferative effects, isoliensinine was found to be the most potent phytochemical. Isoliensinine induced cell cycle arrest at the G1 phase, which possibly occurred by downregulating cyclins and upregulating a cyclin-dependent kinase inhibitor, p21. In addition, isoliensinine also induced cell death by inducing apoptosis mediated by induction of reactive oxygen species (ROS) and alterations of Bcl-associated X protein (Bax), B cell lymphoma 2 (Bcl-2), caspase-3, and poly (ADP-ribose) polymerase-1 (PARP-1). Finally, isoliensinine activated both p38 MAPK and c-Jun N-terminal kinase (JNK) signaling, contributing to apoptosis induction [84]. 

The anticancer effects of neferine, another bisbenzylisoquinline alkaloid found abundantly in the *N. nucifera* plant, have been investigated by many researchers. Yang et al. [85] reported that neferine decreased MCF-7 cell viability; however, a combination treatment of neferine and dehydroepiandrosterone (DHEA), an endogenous steroid hormone, exerted a greater growth-inhibitory effect. Neferine enhanced the anticancer effect of DHEA to induce apoptosis mediated by increased levels of pro-apoptotic factors, such as caspase-3, caspase-8, caspase-9, Bax, p53, p21, transcription factor E2F1, Fas, Fas ligand (FasL), and decreased levels of antiapoptotic factors, such as Bcl-2, B-cell lymphoma-extra-large (Bcl-xL), human inhibitor of apoptosis protein-1 (HIAP-1), HIAP-2, and survivin. From the green seed embryos of *N. nucifera*, Kadioglu et al. [86] isolated neferine and compared its anticancer effect against taxol- and doxorubicin-resistant as well as sensitive MCF-7 cells. Neferine induced cytotoxicity of multidrug-resistant MCF-7 cells to a greater extent compared to the effect on sensitive cell lines by increasing rhodamine 123 (R123) uptake and inhibiting P-glycoprotein (P-gp). In a follow-up study, Law et al. [87] also reported that neferine induced cytotoxicity in drug-resistant and -sensitive MCF-7 cells. Interestingly, autophagic cell death occurred in apoptosis-resistant cells via green fluorescent protein (GFP)-light-chain 3 (LC3) puncta formation and ryanodine receptor (Ryr) activation, which subsequently activated the calcium-dependent kinase (caMKKβ), leading to the activation of the 5′-adenosine monophosphate-activated protein kinase (AMPK)-mammalian target of rapamycin (mTOR) signaling pathway. In another study, Liu et al. [88] investigated the anticancer effects of neferine on MDA-MB-231 human breast cancer cells. Neferine significantly reduced cell proliferation, migration and invasion as well as increased apoptosis via regulation of the expression of miR-374a and fibroblast growth factor receptor 2. Neferine also suppressed the activation of PI3K, Akt, MEK, and ERK signaling pathways.

Similar to neferine, liensinine and nuciferine are two other active bisbenzylisoquinoline alkaloids found in the *N. nucifera* plant. Kang et al. [89] treated MCF-7 and MDA-MB-231 breast cancer cells with liensinine and nuciferine to investigate their anticancer properties. Liensinine and nuciferine both decreased cell viability and inhibited cell proliferation via apoptosis induction and cell cycle arrest in a concentration-dependent manner. Apoptosis was mediated by a mitochondrial-dependent pathway by decreasing Bcl-2/Bax ratio, activating caspase-3, and activating caspase-mediated PARP cleavage. Liensinine was found to be more potent in inhibiting proliferation, invasion and migration in both cell lines compared to nuciferine. In addition, both compounds inhibited osteoclast differentiation and bone resorption by blocking the receptor activator of NF-κB ligand and decreasing the secretion of cathepsin K and MMP-9. Additionally, the anticancer effects of liensinine have been studied by Zhou et al. [90], and the mechanistic results are rather unique. It was determined that treatment of MDA-MB-231 and MCF-7 human breast cancer cells with liensinine reduced the viability of these cells. Mechanistically, liensinine inhibited autophagy and induced the accumulation of autophagosomes and lysosomes by inhibiting their fusion via the recruitment of Ras-related protein Rab-7a (RAB7A) to lysosomes. This inhibition of autophagy/mitophagy by liensinine led to the sensitization of doxorubicin-induced cell death of breast cancer cells via increased dynamin-1-like protein (DNM1L) dephosphorylation and translocation to mitochondria and increased mitochondrial fission. 

Very recently, Huang et al. [91] isolated three new pairs of benzyltetrahydroisoquinoline alkaloid epimers, seco-neferine A-F, from an ethanolic extract of plumula nelumbinis, a commonly used health food and traditional Chinese medicine. All six compounds were tested against MDA-MB-231 human breast cancer cells, and seco-neferine F displayed a maximum but moderate cytotoxic effect. The mechanisms behind the observed anti-breast cancer effect have not been explored.

There are at least three studies that investigated the anticancer potential of *N. nucifera* using in vivo breast cancer models. In addition to the in vitro studies presented earlier, Yang et al. [79] evaluated the effects of NLE on nude mice with an MCF-7 tumor xenograft and found that it effectively reduced tumor volume and tumor weight. Mechanistically, it was demonstrated that NLE can suppress intratumor expression of human epidermal growth factor receptor 2 (HER2), phospho-HER2 (p-HER2), and Fas (Table 3). Chang et al. [80] studied the effects of NLE in female C57BL/6 nude mice xenografted with MDA-MB-231 cells treated with various concentrations of NLE (0.5, 1, and 2%). There was a significant reduction of tumor size in animals receiving NLE-treated cells compared to control. Immunohistochemical staining of tumor tissue with CD31 antibody revealed NLE-mediated inhibition of angiogenesis as demonstrated by CD31. However, mechanisms of action of the observed antitumor and antiangiogenic effects were not studied in vivo. In addition to the in vitro experiment presented earlier, Zhou et al. [90] further performed in vivo studies to replicate the effects of liensinine on the autophagy of breast cancer cells. In this study, 5–7-week-old female nude mice were inoculated with MDA-MB-231 cells and treated with either liensinine (60 mg/kg) or doxorubicin (2 mg/kg) alone or a combination of both. It was determined that the combination treatment of liensinine and doxorubicin significantly reduced tumor growth. Mechanistically, liensinine suppressed autophagy/mitophagy, increased mitochondrial translocation of DNM1L, and increased the colocalization of DNM1L and translocase of outer mitochondrial membrane 20 (TOMM20). All of these mechanisms led to the sensitization of doxorubicin-induced cell death via the promotion of DNM1L-mediated mitochondrial fission. 

#### 3.2.2. Cervical Cancer

*N. nucifera*’s anticancer effect on cervical cancer has gained recent research interest. Currently, there are at least three studies that have explored this area. Maneenet et al. [67] reported a considerable cytotoxic activity of an ethanolic extract of *N. nucifera* petals against HeLa human cervical cancer cells with a preferential cytotoxicity (PC_50_) of 10.9 µM. Phytochemical characterization of the extract revealed the presence of nine benzylisoquinoline alkaloids which exhibited significant antiausterity activity against HeLa cells. One compound, namely (−)-lirinindine, showed the greatest cytotoxicity as evidenced by cell shrinkage, plasma blebbing, and ultimately total cell death within 10 h. Mechanistically, the cytotoxic effect was achieved via induction of apoptosis, activation of caspase-3, decreased levels of Bcl-2, and downregulation of the Akt/mTOR pathway. Li et al. [92] studied the anticancer effects of liensinine on Caski, C33A, HeLa, and SiHa human cervical cancer cells. The alkaloid inhibited the proliferation of all cervical cancer cells and suppressed the colony formation of C33A and HeLa cells. A mechanistic study indicated that liensinine registered cell cycle arrest at the G0/G1 phase, increased apoptosis, elevated the levels of caspase-9 and p21 and decreased the expression of Mcl-1, CDK2, p-Akt, and GSK3α. Dasari et al. [93] focused on researching the effects of neferine on HeLa and SiHa cervical cancer cells. Treatment with various concentrations of neferine suppressed viability, induced cytotoxicity, and reduced migration in a concentration-dependent manner. Neferine induced apoptosis by promoting excess ROS formation, increasing oxidative damage, activating the expression of a DNA damage response marker (pH2AX) and upregulating the levels of pro-apoptotic markers, such as cytochrome c, Bax, cleaved caspase-3, caspase-9 and cleaved PARP1, while downregulating anti-apoptotic factors, such as Bcl-2, translational controlled tumor protein (TCTP), procaspase-3, and procaspase-9. Neferine was also shown to induce autophagy by promoting the conversion of LC3 protein from LC3-I to LC3-II and by increasing the formation of autophagosomes which was determined by an increase in autophagy factors, such as Beclin1, atg-4, atg-5, and atg-12. Migration of the cancer cells was also reduced due to cell cycle arrest at the G1/G0 phase of the cell cycle. 

#### 3.2.3. Colon Cancer

To investigate the anticancer properties of an ethanol crude extract from the stamen (a part of a flower) of Ba lotus (*N. nucifera*), Zhao et al. [94]) incubated HCT-166 human colon carcinoma cells with the extract. The test material showed an antiproliferative effect by inducing apoptosis mediated by downregulation of Bcl-2, Bcl-xL, inducible nitric oxide synthase (iNOS), cyclooxygenase-2 (COX-2), NF-κB, and upregulation of caspase-3, caspase-8, caspase-9, Fas, FasL, tumor necrosis factor-related apoptosis-inducing ligand (TRAIL), death receptor 4 (DR4), death receptor 5 (DR5), and IκBα. Additionally, the extract reduced matrix degradation and metastasis by increasing the expression of TIMP-1 and TIMP-2, which inhibited the activation of MMP-2 and MMP-9. Kadioglu et al. [86] evaluated the effect of neferine against HCT-8 human colon cancer cells. Neferine reduced cell viability by inhibiting P-gp and increasing the uptake of R123, a substrate of P-gp. Manogaran et al. [95] studied the effect of neferine and isoliensinine on HCT-15 human colon cancer cells and found that each alkaloid reduced cell viability. The underlying mechanisms for the anticancer effects of these compounds included the induction of apoptosis by increasing levels of ROS, membrane permeability, Bax, caspase-3, caspase-9, cleaved PARP, intracellular calcium, and decreasing the mitochondrial membrane potential (∆ψM) and expression of Bcl-2. In addition, Western blot analysis showed that both compounds induced the activation of the MAPK pathway and increased the protein expression of p38. Qi et al. [96] treated HT29 and HCT116 human colon cancer cells and CT26 murine undifferentiated colon carcinoma cells with nuciferine. They found that the alkaloid reduced cell viability, inhibited cell proliferation and suppressed cell invasion. In CT26 cells, nuciferine also decreased the expression of PI3K, interleukin-1β (IL-1β), and p-Akt. In another study, liensinine, extracted from the seed embryo of *N. nucifera,* suppressed cell proliferation in HT29 and DLD-1 human colorectal cancer cells. Mechanistic studies revealed that liensinine increased apoptosis, expression levels of cleaved caspase-3, cleaved PARP, cell cycle arrest in the second growth and mitosis (G2/M) phase, phosphorylated CDK1 1 (p-CDK1), cyclin A2, phosphorylated JNK (p-JNK), and Bax, and decreased Bcl-2 and Bcl-xL [97]. Guon and Chung [98] isolated two flavonol glycosides, namely hyperoside and rutin, from the roots of *N. nucifera* and determined their anticancer properties using HT29 human colorectal cancer cells. The two flavonols exhibited cytotoxicity, reduced cell viability, and inhibited cell proliferation via increased apoptosis. The observed apoptosis-inducing activity was mediated by increased Bax, caspase-3, caspase-8, and caspase-9 levels, an elevated Bax/Bcl-2 ratio, and a decreased level of Bcl-2. 

Qi et al. [96] observed the anticancer properties of nuciferine through an in vivo approach in addition to the in vitro studies mentioned previously. Nude mice were subcutaneously implanted with CT29 cells and injected with 9.5 mg/kg nuciferine 3 times a week for 3 weeks immediately following tumor cell implantation or when tumor xenografts reached a size of 100 mm^3^. The alkaloid resulted in reduced tumor weights in both the treatment groups compared to the control. However, the antitumor mechanisms were not explored in vivo. Wang et al. [97] also investigated the anticancer properties of liensinine in vivo by feeding it to 6–8-week-old female BALB/c nude mice injected with HT29 human colorectal cancer cells. Liensinine suppressed colorectal tumorigenesis and reduced tumor volume by decreasing cell proliferation and the expression of Ki-67.

#### 3.2.4. Esophageal Cancer

There has been a limited amount of research investigating the anticancer properties of *N. nucifera* in esophageal cancer. An et al. [99] treated KYSE30, KYSE150, and KYSE510 human esophageal squamous cell carcinoma cells with neferine. They found that the alkaloid suppressed cell proliferation in all three cell lines and suppressed colony formation of KYSE30 and KYSE150 cells. The mechanistic studies revealed that neferine induced apoptosis, arrested cells at the G2/M, triggered the accumulation of ROS, increased the expression of cleaved PARP, cleaved caspase-3, cleaved caspase-9, p21, p-JNK, and decreased the expression of Bcl-2, cyclin B1, and Nrf2.

#### 3.2.5. Eye Cancer 

There is at least one report of the *N. nucifera* alkaloid neferine on retinoblastoma, a common primary intraocular childhood and infant malignancy. Wang et al. [100] observed that neferine inhibited the proliferation and viability of the WERI-Rb-1 human retinoblastoma cell line associated with reduced protein expression of Ki-67 and Survivin. Neferine treatment reduced microtubule-like structure formation, the number of nodes/high power field, and VEGF protein levels, demonstrating its anti-invasive capability. The results also suggested that neferine caused mitochondrial dysfunction in retinoblastoma cells through decreased SOD and GSH levels, increased MDA content, and apoptosis through downregulation of Bcl-2 and c-Myc, upregulation of Bax expression and cleavage of caspase-3 and caspase-9.

In addition to the in vitro study, Wang et al. [100] investigated the anticancer effects of neferine using an in vivo retinoblastoma model. WERI-Rb-1 cells were subcutaneously injected in female athymic nude mice, and the tumor-bearing animals were treated with 0.5–2 mg/kg of neferine every 3 days for 30 days. Results showed a decrease in tumor volume and weight mechanistically through reduced expression of Ki-67 and VEGF, decreased SOD activity, and increased MDA content.

#### 3.2.6. Gallbladder Cancer

The anticancer effects of *N. nucifera* have not been thoroughly investigated in regard to gallbladder cancer. There is only one study by Shen et al. [101] that explored the effects of liensinine on GBC-SD and NOZ human gallbladder carcinoma cells. Treatment with various concentrations of liensinine inhibited proliferation and suppressed cancer cell growth by inducing apoptosis and arresting the cells in the G2/M phase of the cell cycle. Western blot analysis evidenced an upregulation of cleaved caspase-3, caspase-9, PARP and Bax alongside th downregulation of Bcl-2, cyclin B1, CDK1, CDC25C, zinc finger X-chromosomal protein (ZFX), PI3K and p-Akt.

Shen et al. (2019) also performed an in vivo study on BALB/c nude mice which were injected with NOZ cells. The tumor-bearing mice were treated with 2 mg/kg of liensinine intraperitoneally. Liensinine was found to reduce tumor volume and weight with a parallel decrease in intratumor Ki-67 expression.

#### 3.2.7. Gastric Cancer

MFC human gastric cancer cells were treated with water-soluble polysaccharides extracted from *N. nucifera* seeds, and the results showed inhibition of cell proliferation [102]. However, the underlying mechanism of action was not reported. Huang et al. [103] investigated neferine’s involvement in the reversal of adriamycin (ADM) resistance in SGC7901/ADM human gastric cancer cells. It has been found that the combined treatment of hyperthermia and neferine can reverse the multidrug resistance of ADM in human gastric cancer cells by increasing cell membrane fluidity, accelerating the passive ADM permeation, and decreasing P-gp expression and the levels of multidrug resistance 1 (MDR-1) mRNA. Xue et al. [104] treated GIST-1 human gastrointestinal stromal tumor cells and SGC7901 human gastric cancer cells with neferine. In the GIST-1 cell line, neferine inhibited cell viability, proliferation, and migration. The anticancer mechanism involved the induction of apoptosis, increased expression of p15, p16, p21, Bax, cleaved caspase-3, cleaved caspase-9, microRNA-449a (miR-449a), and decreased expression of cyclin D1, Bcl-2, MMP-2, MMP-9, p-PI3K, p-Akt, Notch1, Notch2, and Notch3. In the SGC7901 cell line, neferine suppressed cell migration through increasing apoptosis and the protein expressions of Bax, cleaved caspase-3, cleaved caspase-9, and miR-449a. The alkaloid also downregulated Bcl-2, MMP-2, and MMP-9 in the SGC7901 cell line. In another study, various concentrations of 7-hydroxydehydronuciferine, a compound isolated from a methanolic extract of the leaves of *N. nucifera* Gaertn. cv. *Rosa-plena*, inhibited the proliferation of AGS human gastric cancer cells possibly by antioxidant activity [105]. Yang et al. [106] studied the effect of liensinine, an alkaloid extracted from the seeds of *N. nucifera* Gaertn., using BGC823 and SGC7901 human gastric cancer cells. Liensinine inhibited cell proliferation by increasing apoptosis, the expression of cleaved caspase-3, cleaved caspase-9, cleaved PARP, Bax, and ROS, and decreasing the expression of p-Akt and Bcl-2. The alkaloid also increased G0/G1 cell arrest and decreased levels of cyclin D1 and CDK4. 

One in vivo study has been conducted regarding the anticancer effects of liensinine in gastric cancer. Yang et al. [106] tested BALB/c homozygous nude mice, xenografted with SGC7901 human gastric cancer cells, by injecting them with liensinine every two days for a month. Liensinine showed anticancer properties by reducing tumorigenesis and inhibiting cell proliferation. Mechanistic studies based on immunohistochemical analysis of tumor tissues revealed that liensinine decreased the expression of Ki-67, a marker for cell proliferation.

#### 3.2.8. Head and Neck Cancers

Neferine’s effects on mitigating head and neck cancer have been recently explored by Zhu et al. [107]. In this in vitro study, neferine reduced viability, inhibited proliferation and suppressed migration in 3 different head and neck cancer cell lines, namely HN6 and CAL27 tongue squamous cell carcinoma and HN30 pharyngeal squamous cell carcinoma. Neferine was found to trigger apoptosis and autophagy as well as increase the levels of ROS in the HN30 and Cal 20 cells. Neferine also led to the activation of the apoptosis signal-regulating kinase 1 (ASK1)/JNK pathway, leading to apoptosis. In addition, neferine induced autophagy by promoting the generation of autophagosomes, while inhibiting autophagic influx. This was observed by an increased conversion of LC3-I to LC3-II and the accumulation of p62. 

In the in vivo study, CAL27 cells were injected in 5-week-old male BALB/c nude mice, which were then treated with 10 mg/kg neferine. Neferine reduced tumor volumes by increasing apoptosis and pro-apoptotic factors, such as cleaved caspase-2 and cleaved PARP. In addition, an increase in autophagy was observed, as evidenced by an increase in the conversion of LC3-I to LC3-II conversion and p62 levels [107].

#### 3.2.9. Hematological Cancers 

Limited information is available on the anticancer effect of *N. nucifera* in hematological cancers. One study used a pure compound, kaempferol, which was isolated from a methanolic extract of *N. nucifera* stamens. It has been found that kaempferol was not cytotoxic to KU812F cells, a human basophilic cell line derived from chronic myelocytic leukemia. However, kaempferol suppressed the expression of FcεR1 on the cell surface, decreased the mRNA levels of FcεR1 α- and γ-chains, reduced intracellular Ca^2+^ concentration, and diminished FcεR1-mediated histamine release in KU812F cells [108]. In a separate study, Qin and co-workers [109] experimented with neferine as a possible treatment for multi-drug resistant leukemia. They tested neferine’s effect against imatinib-resistant K562/G01 human myelogenous leukemia cells. It was found that neferine, at high concentrations of 16 µM or greater, reduced cell survival rate in a concentration-dependent manner. Lower concentrations of neferine did not show any effects on cell survival. Further experimentation revealed that neferine could reverse imatinib resistance and increase its concentrations in STI571-resistant K562/G01 cells by downregulating the expression of P-gp and *MDR-1* mRNA.

#### 3.2.10. Laryngeal Cancer

Tripartite motif-containing 44 (TRIM44) was found to be upregulated in 4 human laryngeal squamous cell carcinoma (LSCC) cell lines (AMC-HN-8, HEP-2, TU-212, and TU-686), and the knockdown of TRIM44 led to suppressed cell growth. In a recent study, 2 of these cell lines, AMC-HN-8 and TU-212, were treated with nuciferine for 24 h. Nuciferine inhibited cell survival by reducing TRIM44 expression, which led to decreased levels of Toll-like receptor 4 (TLR4) and the suppression of Akt signaling. However, with the use of a TRIM44-overexpressing plasmid transfected into LSCC cells, it was found that the overexpression of TRIM44 offset the anticancer properties of nuciferine [110]. 

#### 3.2.11. Liver Cancer 

Zheng et al. [102] treated HuH-7 human liver cancer and H22 mouse hepatocarcinoma cells with water-soluble polysaccharides extracted from *N. nucifera* seeds for 48 h, and the results showed inhibition of cell proliferation with no reported underlying mechanism of action. An ethanolic extract of the seedpod of *N. nucifera* showed antiproliferative and cytotoxic properties in HepG2 human liver cancer cells via antioxidant activity as demonstrated by the inhibition of the peroxidation of linoleic acid. Phytochemical analysis revealed the presence of pure compounds, such as catechin, kaempferol, quercetin, and hyperoside, in the seed extract [111]. Duan et al. [112] investigated the anticancer effects of procyanidins found in the water-acetone extract from the seed pods of *N. nucifera*. It was found that the procyanidins decreased viability, caused DNA damage and S phase cell cycle arrest, and induced autophagy of HepG2 cells by decreasing the mitochondrial membrane potential and increasing the levels of LC3, GFP-LC3, and ROS.

Multiple studies investigated the anti-liver cancer properties of neferine, an alkaloid isolated from *N. nucifera*. In one study, Xiao-Hong et al. [113] found that neferine reversed the thermotolerance of HepG2 human hepatocarcinoma cells by increasing apoptosis and downregulating the expression of Bcl-2. Yoon et al. [114] used Hep3B human liver cancer cells and reported that neferine treatment induced growth inhibition and decreased cell viability. The mechanisms associated with these anticancer effects included increased apoptosis, elevated expression of Bim, Bid, Bax, Bak, Puma, caspase-3, caspase-6, caspase-7, caspase-8, and PARP, and decreased expression of CDK4, E2F transcription factor 1 (E2F1), cyclin D1, cyclin D3, and cellular myelocytomatosis (c-Myc) oncogene. In another study, neferine reduced cell viability and induced ROS-mediated apoptosis in HepG2 human liver cancer cells. Additionally, the alkaloid increased the protein expression of Bax, Bad, cleaved caspase-3, caspase-9, PARP, tumor necrosis factor-α (TNF-α), and decreased Bcl-2. Finally, neferine increased the levels of phosphatase and tensin homolog (PTEN), p53, p38, ERK1/2, MAPK, and decreased p-Akt [115]. Deng et al. [116] also used HepG2 human liver cancer cells and found that neferine exhibited cytotoxicity and suppressed cell migration and invasion by increasing apoptosis and protein expression of E-cadherin and decreasing the expression of vimentin, snail, and N-cadherin. When both HepG2 and Hep3B human liver cancer cell lines were treated with neferine, Law et al. [87] ascertained that the alkaloid induced autophagy and displayed cytotoxicity by activating various signaling cascades, such as Unc-51-like autophagy activating kinase-1-protein kinase RNA-like ER kinase (Ulk-1-PERK) and AMPK-mTOR and increasing the amount of intracellular calcium via Ryr activation. 

Two studies examined the anticancer effects of isoliensinine, an alkaloid derived from *N. nucifera* embryos, on hepatocellular cancer in vitro. Shu et al. [117] treated Hep2G, Huh-7 and H22 hepatocellular carcinoma cells with various concentrations of isoliensinine and found that it suppressed proliferation and induced apoptosis through a decrease in NF-κB activity in HCC cells, as well as through decreased phosphorylation of the NF-κB p65 subunit. The researchers also observed a decrease in NF-κB target proteins, such as Bcl-2, Bcl-xL and MMP-9, and an increase in caspase-3. In an extension of the previous study, Shu et al. [118] observed that isoliensinine suppressed NF-κB in liver cancer cells through impairing protein phosphatase 2A (PP2A)/inhibitor of PP2A (I2PP2A) interaction and stimulating PP2A-dependent p65 dephosphorylation at serine536. 

Several studies have also explored the anticancer properties of *N. nucifera* using in vivo liver cancer models. In one such study, H22 human hepatocellular carcinoma cells were inoculated in 6–8-week-old female Kunming mice. After 24 h, the animals were treated with water-soluble polysaccharides from the seeds of *N. nucifera* (LSPS). After 14 days of LSPS treatment, reduced tumorigenesis was observed, as there was a significant decrease in tumor weight. Mechanistic studies revealed reduced malondialdehyde (MDA) levels in the liver, increased superoxide dismutase (SOD) activity, and an increased spleen and thymus index. In addition, there were increased levels of TNF-α and IL-2 [102]. In another study, Horng et al. [119] included 0.5–2.0% of an aqueous NLE, containing phenolic compounds such as gallic acid, catechin, peltatoside, rutin, isoquercitrin, miquelianin, and astragalin, in the diet of male Sprague-Dawley rats with hepatic carcinoma induced by diethylnitrosamine (DEN) via drinking water. By the assessment of a histopathologist, the group of rats fed DEN+2.0% NLE showed a reduction of liver carcinomas by 30% compared to the control group (DEN-fed rats). Furthermore, NLE reduced tumor size, decreased hepatocellular damage and reduced the number of glutathione S-transferase pi (GSTπ)-positive cells. Additional studies showed that NLE elicited an antioxidant effect as characterized by decreased lipid peroxidation, increased glutathione (GSH), glutathione peroxidase (GPx), SOD and catalase (CAT) as well as reduced levels of alanine aminotransferase (ALT), aspartate transaminase (AST), and albumin. Finally, NLE decreased the expression of Ras-related C3 botulinum toxin substrate 1 (Rac1), PKCα, GST, TNF-α, and IL-6. Yang et al. [120] also used dietary NLE to determine its anticancer properties in 4–5-week-old Wistar rats with 2-acetylaminofluorene (2-AAF)-induced hepatocarcinogenesis. After six months of treatment, 0.5–2% NLE was found to reduce 2-AAF-induced hepatic fibrosis (appears during the development of premalignant lesions) compared to the control (2-AAF-treatment-only rats), lower hepatic fibrosis, reduce liver weight, and enhance antioxidative potential by decreasing lipid peroxidation, levels of triglycerides, total cholesterol, α-fetoprotein (AFP), IL-6, TNF-α, AST, ALT, γ-glutamyl transferase (γGT), GSTπ, and 8-hydroxy-2′-deoxyguanosine (8-OHdG). NLE also led to increased levels of nuclear factor erythroid 2-related factor 2 (Nrf2), CAT, GPx, and SOD-1. Deng et al. [116] injected HepG2 and Bel-7402 human hepatocellular carcinoma cells into 4-week-old male BALB/c mice and treated them with oxaliplatin (a drug used for the treatment for hepatocellular carcinoma) in the presence or absence of neferine. Neferine potentiated the tumor volume-reducing activity of oxaliplatin by depressing cell proliferation based on Ki-67 expression. The combination of oxaliplatin and neferine increased the expression of E-cadherin and decreased the expression of vimentin compared to oxaliplatin treatment alone. 

There were two similarly designed studies in which the antineoplastic effects of isoliensinine were studied in vivo. H22 hepatocellular cancer cells were inoculated in male Kunming mice, and athymic male nude mice were injected with Huh-7 cells. The Kunming mice were split into 3 groups and treated with 3 or 10 mg/kg/day isoliensinine for 10 days, intraperitoneally or by gavage, or with normal saline. Athymic nude mice were also treated with isoliensinine, intraperitoneally, at a dose of 3 or 10 mg/kg/day for 3 weeks. In both models, isoliensinine showed reduced tumor growth through the apoptosis of tumor cells mediated by increased caspase-3 activity, decreased levels of Bcl-2, Bcl-xL, and MMP-9, and decreased NF-κB p65 phosphorylation [117]. In a follow-up study, the same research group [118] found that oral administration of isoliensinine at 10 mg/kg once daily for 10 days reduced tumor growth in Huh-7 xenograft model containing I2PP2A via increased caspase-3 activity. 

#### 3.2.12. Lung Cancer 

There are many studies evaluating the anticancer properties of *N. nucifera* in lung cancer. One study treated A549 and H460 human non-small cell lung cancer cell lines with *N. nucifera* ethanolic seed pod extract and found it to reduce cell proliferation and colony formation. The extract also induced apoptosis through increased cleavage of PARP and increased phosphorylation of H2A histone family member X (γ-H2AX). Additionally, the extract-treated cells showed the ability to reduce the expression of mRNA and protein of Axl in both cell lines, indicating that Axl is a novel therapeutic target of the antiproliferative and proapoptotic activities of *N. nucifera* [121]. 

Neferine, a major bisbenzyliso-quinoline alkaloid component of *N. nucifera*, has been widely studied for its anticancer effects in lung cancer. Poornima et al. [122] investigated the effects of neferine on A549 human lung adenocarcinoma cells and found that it markedly suppressed cell proliferation in a concentration-dependent fashion. Neferine also induced autophagy as indicated by acidic vesicular accumulation and autophagosome formation measured through the expression of microtubule-associated protein-1 LC3B and conversion of LC3-I to LC3-II. Neferine decreased the expression of PI3K, Akt and mTOR, resulting in the induction of autophagy through mTOR inhibition. Finally, the investigators found that neferine-induced autophagy was mediated through ROS generation and subsequent depletion of intracellular GSH levels. In a subsequent study, the same research group [123] further investigated the effects of neferine on a panel of human lung cancer cells, such A549, H520, H661, and H441. The nontoxic nature of neferine was established in BEAS-2B normal lung cells. The cumulative findings suggested that neferine induced cytotoxicity in all lung cancer cells. In A549 cells, neferine induced apoptosis through inhibition of Bcl-2 expression and increased expression of Bax, Bad, and cytochrome c (cyt c), while the mitochondrial membrane potential was decreased. The mechanism of the observed anticancer effect of neferine was established through upregulation of p53, p27, and PTEN and downregulation of cyclin D1 and NF-κB. Increased ROS production, depletion of cellular antioxidants and activation of JNK, ERK 1/2, and p38 MAPK were also observed. 

Kadioglu et al. [86] used neferine to study its cytotoxicity against paclitaxel (an anticancer drug)-resistant and -sensitive A549 cell lines. The study showed that neferine was more cytotoxic to the paclitaxel-resistant cell line. Neferine inhibited P-gp based on increased R123 uptake in the paclitaxel-resistant cell line in a comparable manner to verapamil, a known P-gp inhibitor. These results suggest that neferine has potential as a drug candidate for the treatment of multidrug-resistant cancers, since the overexpression of P-gp is linked to the efflux of chemotherapeutic drugs. Law et al. [87] confirmed the cytotoxic effect of neferine against A549 cells and also observed similar effects on other lung cancer cell lines, such as H1299 and LLC-1. Colony formation in H1299 cells was also inhibited by neferine. Neferine induced autophagy in apoptosis-resistant cell lines (A549 and LLC-1). Neferine also induced GFP-LC3 puncta formation, indicating apoptotic activity via autophagosome formation. 

Kalai Selvi et al. [124] found that neferine induced cytotoxicity and enhanced the therapeutic effect of cisplatin through stimulation of non-canonical autophagy and ROS generation in A549 cells. This mechanistic study found that the neferine-cisplatin combination induced autophagy mediated through the hypergeneration of ROS, GSH depletion, and downregulation of the PI3K/Akt/mTOR pathway. In neferine- and cisplatin-treated A549 cells, autophagosome formation was measured through the expression of LC3B and increased conversion of LC3-I to LC3-II. The same research group extended the previous study and showed that neferine potentiated the cytotoxic effects of cisplatin. Combined treatment also exhibited apoptosis induction through sub-G1 cell cycle arrest and increased Bax, Bad, Bak, cyt c, caspase-3, caspase-9, PARP cleavage, and ROS generation. Additionally, the neferine and cisplatin combination downregulated the protein levels of focal adhesion kinase (FAK) and VEGF and suppressed the activity of MMP-2 [125].

Liu et al. [126] investigated the effect of nuciferine, another phytochemical present in *N. nucifera*, against nicotine (an addictive component of tobacco)-induced tumor promotion in A549 cells. Nuciferine was found to inhibit the proliferation of A549 cells in the presence of nicotine and suppressed the migration and invasion of tumor cells in the presence or absence of nicotine. Nicotine-activated Wnt/β-catenin signaling was substantially reduced after treatment with nuciferine, which was achieved by the stabilization and increased expression of Axin, while no change in adenomatous polyposis coli and glycogen synthase kinase-3β in nuciferine-treated cells was observed. Wnt/β-catenin signaling target proteins, such as c-myc, cyclin D, and VEGF-A, showed decreased expression when treated with nuciferine. Qi et al. [96] treated A549 and NCI-H1650 human lung adenocarcinoma cells with nuciferine to determine its anticancer potential. The results showed that nuciferine reduced cell viability and suppressed cell invasion via unknown mechanisms.

There are three in vivo studies that evaluated the anticancer effects of *N. nucifera* on lung cancer. Wu et al. [81] examined the anti-metastatic effects of NLE and NLPE in primary lung tumors developed following orthotopic implantation of 4T-1 cells in BALB/c mice. The researchers observed fewer lung micrometastases in the 1% NLE- and 0.25% NLPE-treated groups compared to control groups. The weight of the lungs of the treated group was also reduced compared to the control groups. Based on immunohistochemical analysis, NLE and NLPE reduced the activation of protein kinase C α in the lung tissue. Liu et al. [126] explored the antitumor effects of nuciferine (50 mg/kg 3 times a week for about 3 weeks) in BALB/c mice injected with A549 tumor cells in the presence or absence of nicotine exposure. The results indicated that nuciferine registered a significant reduction of tumor size and weight in the presence of nicotine. Mechanistic study revealed that nuciferine enhanced apoptosis of tumor cells, upregulated the expression of Bax and Axin, and downregulated the expression of Bcl-2 and β-catenin. Compared to the nicotine control group, the nuciferine-treated groups did not show any reduction in body weight or decline in liver and kidney functions. Sivalingam et al. [127] used diethylnitrosamine (DEN) to induce lung carcinogenesis in albino male Wistar rats. The rats were then treated with 10–20 mg/kg of neferine by oral intubation for 20 alternate days. The investigators found that neferine decreased lung weight and reduced pathological damage in DEN-induced rats. Neferine was shown to reduce oxidative stress in DEN-induced lung carcinogenesis through decreased ROS generation, lipid peroxidation and protein carbonyl production, as well as increased GSH, SOD, GST, and CAT. The DEN-induced increase in glycoproteins in lung tissue was significantly decreased after neferine treatment. Neferine enhanced mitochondrial-mediated apoptosis through a decreased expression of Bcl-2 and increased expression of p53, Bax, caspase-9 and caspase-3. DEN-induced animals exhibited increased inflammatory genes and proteins, such as NF-κB, COX-2, CYP2E1 and VEGF, as well as increased expression of PI3K, Akt and mTOR genes, which were all decreased with neferine treatment.

**Table 2 cancers-14-00529-t002:** Potential anticancer effects and mechanisms of action of *N. nucifera*-derived constituents based on in vitro studies.

Materials Tested	Cell Lines Used	Conc. (Duratdion)	Anticancer Effects	Mechanisms	References
*Breast cancer*					
Aqueous rhizome extract	MDA-MB-231 (human breast cancer)	1–1000 µg/mL (24 h)	Inhibited proliferation and migration	↓MMP-2; ↓MMP-9	Karki et al., 2008 [78]
Flavonoid-rich leaf extract	MCF-7 (human breast cancer)	0.5–3 mg/mL (24 h)	Inhibited proliferation	↑p16; ↑p21; ↑p27; ↓cyclin E; ↓cyclin D1; ↓CDK2; ↓CDK4; cell cycle arrest in G0/G1 phase; ↓cyclin D1/CDK4 complex; ↓cyclin E/CDK2 complex; ↑Rb-E2Fcomplex; ↓Fas	Yang et al., 2011 [79]
Aqueous leaf extract	MDA-MB-231 (human breast cancer)	0.5–5 mg/mL (24 h)	Inhibited angiogenesis; inhibited proliferation; decreased migration rate; reduced cell invasion properties	↓MMP2; ↑TIMP; ↓VEGF, ↓CTGF; ↓VEGFR2; ↓NF-κB p65; ↓PI3K-Akt-ERK; ↓RAS; ↓MEK; ↓ERK; ↓Akt	Chang et al., 2016 [80]
Aqueous and methanol leaf extracts	4T-1 (mouse mammary carcinoma); MDA-MB-231 (human breast cancer)	2–4 mg/mL (24 h)	Reduced cell viability and attenuated migration	↑Apoptosis; ↓RhoA; ↓Rac1; ↓Cdc42; ↓ERK; ↓p38; ↓MAPK	Wu et al., 2017 [81]
Methanol and acetone leaf and flower extracts	MCF-7 (human breast cancer)	6.25–100 µg/mL (24 h)	Inhibited proliferation and reduced viability	Not reported	Arjun et al., 2012 [82]
Methanol floral receptacle extract	MCF-7 (human breast cancer)	200–600 µg/mL (24 h)	Induced cytotoxicity	Antioxidant activity	Krubha & Vasan, 2016 [83]
Isoliensinine, liensinine & neferine	MCF-10A (human breast cancer)	5–20 μM (48 h)	Induced cell death	↑Apoptosis; ↑oxidative stress; ↑p38; ↑MAPK; ↑JNK	Zhang et al., 2015 [84]
Neferine	MCF-7 (human breast cancer)	2–20 mg/mL (48 h)	Inhibited proliferation; reduced cell viability	↑Apoptosis; ↑caspase-3; ↑caspase-8; ↑caspase-9; ↑Bax; ↑p53; ↑p21; ↑E2F1; ↑Fas; ↑FasL; ↓Bcl-2; ↓Bcl-xL; ↓HIAP-1; ↓HIAP-2	Yang et al., 2016 [85]
Neferine	MCF-7 (human breast cancer)	0.039–100 μM (72 h)	Induced cytotoxicity	↑R123 uptake; ↓P-gp	Kadioglu et al., 2017 [86]
Neferine	MCF-7 (human breast cancer)	1–5 μM IC_50_ = 41.1 μM	Induced autophagy	↑Ryr; ↑cytosolic Ca^2+^; ↑AMPk-mTOR; ↑GFP-LC3 puncta; ↑CXCR-4; ↑p-PERK; ↑PERK; ↑SQSTM; ↑Ulk-1	Law et al., 2019 [87]
Neferine	MDA-MB-231 (human breast cancer)	2–10 μM (24 h)	Reduced proliferation	↑Apoptosis; ↓miR-374a; ↓PI3K; ↓Akt; ↓MEK; ↓ERK	Liu et al., 2019 [88]
Liensinine and nuciferine	MCF-7; MDA-MB-231 (human breast cancer)	10–60 µM (24 h)	Inhibited proliferation	↑Apoptosis; ↑Bax/Bcl-2; ↑caspase-3	Kang et al., 2017 [89]
Liensinine	MCF-7; MDA-MB-231 (human breast cancer)	20 µM (24 h)	Reduced viability	↓Autophagy; ↓autophagosome-lysosome fusion; ↓recruitment of RAB7A to lysosomes; ↑mitochondrial fission; ↑DNM1L dephosphorylation; ↑DNM1L translocation	Zhou et al., 2015 [90]
Seco-neferine F	MDA-MB-231 (human breast cancer)	IC_50_: 39 µM (48 h)	Induced cytotoxicity	Not reported	Huang et al., 2021 [91]
*Cervical cancer*					
Ethanolic petal extract; (−)-lirinidine	HeLa (human cervical cancer)	PC_50_: 2–11 μM (24 h)	Displayed antiausterity activities	↑Apoptosis; ↑caspase-3; ↓Bcl-2; ↓p-Akt; ↓p-mTOR	Maneenet et al., 2021 [67]
Isoliensinine	Caski, C33A, HeLa, SiHa (human cervical cancer)	5–25 μM (24 h and 48 h)	Inhibited cell proliferation and colony formation	↑Apoptosis; ↑G0/G1 phase arrest; ↑p21; ↑caspase-9; ↓Mcl-1; ↓CDK2; ↓cyclin E; ↓p-Akt; ↓GSK3α	Li et al., 2022 [92]
Neferine	HeLa, SiHa (human cervical cancer)	5–50 μM (48 h)	Suppressed viability; induced cytotoxicity; reduced migration	↑Apoptosis; ↑Bax; ↑cyt c; ↑cleaved-caspase-3; ↑cleaved-caspase-9; ↑PARP-cleavage; ↓Bcl-2; ↓procaspase-3; ↓procaspase-9; ↓TCTP; ↑beclin-1, ↑atg-4, ↑atg-5; ↑atg-12; ↑LC-3 activation; ↑P62/SQSTM1; ↑G1-G0 phase arrest; ↑ROS	Dasari et al., 2019 [93]
*Colon cancer*					
Ethanolic stamen extract	HCT-116 (human colon cancer)	100–400 µg/mL (24 h)	Showed cytotoxic activity	↑Apoptosis; ↑Fas; ↑FasL; ↑TRAIL; ↑DR4; ↑DR5; ↑caspase-3, ↑caspase-8; ↑caspase-9; ↑Bax; ↓Bcl-2; ↓Bcl-xL; ↓MMP-2; ↓MMP-9; ↑TIMP-1; ↑TIMP-2; ↓iNOS; ↓COX-2; ↓NF-κB; ↑IκBα	Zhao et al., 2017 [94]
Neferine	HCT-8 (human colon cancer)	0.039–100 μM (72 h)	Reduced cell viability	↑R123 uptake; ↓P-gp	Kadioglu et al., 2017 [86]
Neferine; isoliensinine	HCT-15 (human colon cancer)	2–12 µM (24 h)	Showed cytotoxicity	↑Apoptosis; ↑ROS; ↑p38; ↑MAPK; ↑Bax; ↑caspase-9; ↑caspase-3; ↑cleaved PARP; ↑membrane permeability; ↓Bcl2; ↑[Ca^2+^]; ↓∆ΨM	Manogaran et al., 2019 [95]
Nuciferine	CT26 (murine colon carcinoma); HT29 and HCT116 (human colon cancer)	0.05–1.0 mg/mL (24 h)	Reduced cell viability and inhibited cell invasion	↓PI3K; ↓IL1B; ↓p-Akt (CT26 cells only)	Qi et al., 2016 [96]
Liensinine	HT29, DLD-1 (human colorectal cancer)	5–20 µM (24–48 h)	Suppressed cell proliferation	↑Apoptosis; ↑cleaved caspase-3; ↑cleaved PARP; ↑G2/M cell arrest; ↑p-CDK1; ↑cyclin A2; ↑p-JNK; ↑Bax; ↓Bcl-2; ↓Bcl-xL	Wang et al., 2018 [97]
Hyperoxide; rutin	HT29 (human colorectal cancer)	100–200 µM (24 h)	Exhibited cytotoxicity, reduced cell viability and inhibited cell proliferation	↑Apoptosis; ↑Bax; ↓Bcl-2; ↑Bax/Bcl-2 ratio; ↑caspase-3; ↑caspase-8; ↑caspase-9	Guon and Chung, 2016 [98]
*Esophageal cancer*					
Neferine	KYSE30, KYSE150 and KYSE510 (human esophageal squamous cell carcinoma)	5–30 µM (24–48 h)	Suppressed cell proliferation and colony formation	↑Apoptosis; ↑G2/M arrest; ↑cleaved PARP; ↑cleaved caspase-3; ↑cleaved caspase-9; ↑p21; ↓cyclin B1; ↓Bcl-2; ↑ROS; ↑p-JNK; ↓Nrf2	An et al., 2020 [99]
*Eye cancer*					
Neferine	WERI-Rb-1 (human retinoblastoma)	0.1 to 200 μM (24 h)	Inhibited cell proliferation, migration, and viability	↓Ki-67; ↓Survivin; ↓microtubule-like structures; ↓nodes/HPF; ↓VEGF; ↓SOD; ↓GSH; ↑MDA; ↓Bcl-2; ↓c-myc; ↑Bax; ↑cleaved caspase-3; ↑cleaved caspase-9	Wang et al. 2020 [100]
*Gallbladder cancer*					
Liensinine	GBC-SD and NOZ (human gallbladder carcinoma)	40–120 μM (24 and 48 h)	Inhibited proliferation and suppressed colony formation	↑Apoptosis; ↑G2/M phase arrest; ↓cyclin B1; ↓CDK1; ↓CDC25C; ↑cleaved-caspase 3; ↑cleaved-caspase 9; ↑cleaved-PARP; ↑Bax; ↓Bcl-2; ↓PI3K; ↓ZFX; ↓p-Akt	Shen et al., 2019 [101]
*Gastric cancer*					
Water-soluble polysaccharides from seeds	MFC (human gastric cancer)	50–200 μg/mL (48 h)	Showed growth inhibition	Not reported	Zheng et al., 2016 [102]
Neferine	Adriamycin resistant SGC7901/ADM (human gastric cancer)	2.5–40 μg/mL (24 h)	Exerted cytotoxicity and reversed drug resistance	↓P-gp expression; ↓*MDR-1* mRNA	Huang et al., 2011 [103]
Neferine	GIST-1 (human gastrointestinal stromal tumor)	1–10 µM (24 h)	Inhibited cell viability, proliferation and migration	↑Apoptosis; ↑p15; ↑p16; ↑p21; ↓cyclinD1; ↑Bax; ↓Bcl-2; ↑cleaved caspase-3; ↑cleaved caspase-9; ↓MMP-2; ↓MMP-9;↑miR-449a; ↓p-PI3K; ↓p-Akt; ↓Notch1; ↓Notch2; ↓Notch3	Xue et al., 2019 [104]
7-Hydroxydehydronuciferine	AGS (human gastric cancer)	IC_50_: 62.9 μM (duration not specified)	Inhibited cell proliferation	Antioxidant activity	Liu et al., 2014 [105]
Liensinine	BGC823 and SGC7901(human gastric cancer)	40–80 µM (48 h)	Inhibited cell proliferation	↑Apoptosis; ↑cleaved caspase-3; ↑cleaved caspase-9; ↑cleaved PARP; ↓p-Akt; ↓Bcl-2; ↑Bax; ↑ROS; ↑G0/G1 cell arrest; ↓cyclin D1; ↓CDK4	Yang et al. 2019 [106]
*Head and neck cancers*					
Neferine	HN6, CAL27 (tongue squamous cell carcinoma), HN30 (pharyngeal squamous cell carcinoma)	7.5–30 µM (24 h)	Reduced cell viability and inhibited colony formation	↑Apoptosis; ↑G1 arrest; ↓Bcl-1; ↑Bax; ↑ROS; ↑autophagosome formation; ↑LC3; ↑p62; ↑p-JNK; ↑p-ASK1	Zhu et al., 2021 [107]
*Hematological cancers*					
Kaempferol (from methanolic stamen extract)	KU81F (chronic myelocytic leukemia)	3.5–35 µM (24 h)	Did not display cytotoxic effect	↓FcεR1 expression; ↓FcεR1 α- and γ-chains; ↓intracellular Ca^2+^; ↓histamine release	Shim et al., 2009 [108]
Neferine	Imatinib-resistant K562/G01 cells (human myelogenous leukemia)	4–64 µM (48 h)	Reduced cell survival rate and reversed drug resistance	↓P-gp expression; ↓*MDR-1* mRNA	Qin et al., 2011 [109]
*Laryngeal cancer*					
Nuciferine	AMC-HN-8, TU-212 (Laryngeal squamous cell carcinoma)	25–100 µM (24 h)	Inhibited cell survival	↓TRIM44; ↓TLR4; ↓Akt signaling	Li et al., 2021 [110]
*Liver cancer*					
Water-soluble polysaccharides from seeds	HuH-7 (human liver cancer); H22 (mouse hepatocarcinoma)	50–200 μg/mL (48 h)	Inhibited cell proliferation	Not reported	Zheng et al., 2016 [102]
Polyphenolic seed extract	HepG2 (human liver cancer)	6.25–50 µg/mL (24–48 h)	Showed cytotoxicity	Antioxidant activity	Shen et al., 2019 [111]
Procyanidins from seedpod extract	HepG2 (human liver cancer)	12.5–400 μg/mL (6–96 h)	Decreased viability and inhibited cell proliferation	↑Autophagy; S phase arrest; ↑LC3; ↑GFP-LC3; ↑ROS	Duan et al., 2016 [112]
Neferine	HepG2 (human hepatocarcinoma)	10–40 µM (24 h)	Reversed thermotolerance of tumor cells	↑Apoptosis; ↓Bcl-2	Xiao-Hong et al., 2007 [113]
Neferine	Hep3B (human liver cancer); Sk-hep-1 (human hepatic adenocarcinoma)	5–30 µM (24 h)	Induced growth inhibition and decreased cell viability	↑Apoptosis; ↓c-Myc; ↓cyclin D1; ↓cyclin D3; ↓CDK4; ↓E2F-1; ↑Bim; ↑Bid; ↑Bax; ↑Bak; ↑Puma; ↑caspase-3; ↑caspase-6; ↑caspase-7; caspase-8; ↑PARP	Yoon et al., 2013 [114]
Neferine	HepG2 (human liver cancer)	2–25 µM (48 h)	Reduced cell viability	↑Apoptosis; ↑Bax; ↑Bad; ↑cleaved caspase-3; ↑caspase-9; ↑PARP; ↓Bcl2; ↑p53, ↑PTEN; ↓p-Akt; ↑TNF-α; ↑p38; ↑ERK1/2 MAP kinases; ↑ROS	Poornima et al., 2013 [115]
Neferine	HepG2 (human liver cancer)	2.5–100 µM (24–48 h)	Exhibited cytotoxicity and suppressed migration and invasion	↑Apoptosis; ↑E-cadherin; ↓Vimentin; ↓Snail; ↓N-cadherin	Deng et al., 2017 [116]
Neferine	HepG2, Hep3B (human liver cancer)	1–5 µM (2 weeks)	Induced cytotoxicity	↑Autophagy; ↑Ryr; ↑cytosolic [Ca^2+^]; ↑Ulk-1-PERK; ↑AMPK-mTOR	Law et al., 2019 [87]
Isoliensinine	HepG2, Huh-7 and H22 (human liver cancer)	3–10 µg/mL (24–48 h)	Inhibited cell proliferation	↑Apoptosis; ↑sub-G1 DNA; ↑caspase-3; ↓Bcl-2; ↓Bcl-xL; ↓MMP-9; ↓NF-κB activity; ↓p65 phosphorylation; ↑p65/PP2A binding	Shu et al., 2015 [117]; Shu et al., 2016 [118]
*Lung cancer*					
Ethanolic seed pod extract	A549, H460 (human non-small cell lung cancer)	10–80 µM (24, 48 h)	Inhibited cell proliferation and colony formation	↑Apoptosis; ↑cleaved PARP; ↑γ-H2AX; ↓Axl	Kim et al., 2021 [121]
Neferine	A-549 (human lung carcinoma)	1–30 µM (12–72 h)	Inhibited cell proliferation	↑Autophagy; ↓PI3K; ↓Akt; ↓mTOR; ↑ROS; ↓GSH	Poornima et al., 2013 [122]
Neferine	A-549, H520, H661, H44 (human lung carcinoma)	1–30 µM (48 h)	Reduced cell viability	↑Apoptosis; G1 cell cycle arrest; ↓Bcl-2; ↑Bax; ↑Bad; ↑cyt c; ↑cleaved caspase-3; ↑cleaved caspase-9; ↓ΔΨM; ↑p53; ↑p27; ↓cyclin D1; ↓NF-κB; ↑PTEN; ↑p-JNK; ↑p-ERK1/2; ↑p-p38; ↓GSH; ↓SOD; ↓CAT; ↓GPx; ↓GST;↑[Ca^2+^]	Poornima et al., 2014 [123]
Neferine	A549 (human lung carcinoma)	0.039–100 μM (72 h)	Reduced cell viability	↑R123 uptake; ↓P-gp	Kadioglu et al., 2017 [86]
Neferine	A549, H1299, LLC-1 (human lung cancer cells)	10 µM (4 h)	Suppressed cell growth	↑Autophagy; ↑LC3-II	Law et al., 2019 [87]
Neferine Neferine+cisplatin	A549 (human lung cancer cell)	10 µM (48 h)	Induced cytotoxicity	↑Autophagy; ↓GSH; ↑ROS; ↑LC3-II; ↓PI3K; ↓Akt; ↓mTOR	Kalai Selvi et al., 2017 [124]
Neferine Neferine + cisplatin	A549 (human lung cancer cell)	10 µM (12–72 h)	Inhibited proliferation and reduced cell viability	↑Apoptosis; ↑LDH leakage; ↑NO release; sub-G1 cell cycle arrest; ↓Bcl-2; ↑Bax; ↑Bad; ↑cyt c; ↑cleaved caspase-3; ↑cleaved caspase-9; ↑cleaved PARP; ↑p53; ↑ROS; ↓FAK; ↓VGEF; ↓MMP-2	Sivalingam et al., 2017 [125]
Nuciferine	A549 (human lung adenocarcinoma)	10–50 µM (24 h)	Exhibited antiproliferative activity and suppressed tumor cell invasion and migration	↑Apoptosis; ↓Bcl-2; ↑Bax; ↓Wnt/β-catenin signaling; ↑Axin; ↓c-myc; ↓cyclin D; ↓VEGF-A	Liu et al., 2015 [126]
Nuciferine	A549 and NCI-H1650 (human lung adenocarcinoma)	0.05–1.0 mg/mL (24 h)	Reduced cell viability and inhibited cell invasion	Not reported	Qi et al., 2016 [96]
*Nasopharyngeal cancer*					
Alkaloids from seeds	CNE-1 (human nasopharyngeal carcinoma)	50–200 µg/mL (24 h)	Reduced cell proliferation	↑Apoptosis; ↑caspase-3; ↑caspase-8; ↑caspase-9; ↑Bax; ↑Fas;↑FasL; ↓Bcl-2; ↓Bcl-xL; ↓NF-κB; ↑IκB-α	Zhao et al., 2016 [128]
*Neural cancer*					
Neferine	IMR32 (human neuroblastoma)	1–30 µM (24 h)	Suppressed proliferation and migration	↑Apoptosis; G2/M phase arrest; caspase-3 cleavage; PARP cleavage; ↑autophagy; ↑LC3-II; ↑Beclin-1; ↓p-FAK; ↓p-S6K1	Pham et al., 2018 [129]
Nuciferine	SY5Y (human neuroblastoma)	0.05–1.0 mg/mL (24 h)	Reduced cell viability and suppressed cell invasion	↓PI3K; ↓IL1B; ↓p-Akt	Qi et al., 2016 [96]
Nuciferine	U87MG, U251 (human glioblastoma)	20–180 µM (24–72 h)	Inhibited cell proliferation, colony formation, mobility, invasion, migration, and epithelial-to-mesenchymal transition	↑Apoptosis; ↑Bax; ↓Bcl-2; ↓HIF1A; ↓VEGFA; ↑G2 cell cycle arrest; ↓Slug; ↓CDC2; ↑cyclin B1; ↓Vimentin; ↓N-cadherin; ↑E-cadherin; ↓SOX2; ↓p-Akt; ↓p-STAT3	Li et al., 2019 [130]
*Ovarian cancer*					
Neferine	A2780, HO8910; SKOV3 (human ovarian cancer)	1–10 µM (24–72 h)	Exhibited cytotoxic and growth-inhibitory effects	G1 cell cycle arrest; ↓cyclin D; ↑p27; ↑p21; ↑apoptosis; ↑autophagy; ↑LC3-II; ↑Atg7; ↓p-p70S6K; ↓p-4EBP1; ↑p-p38 MAPK; ↑p-JNK	Xu et al., 2016 [131]
*Prostate cancer*					
Polyphenolic seed extract	LNCaP (human prostate adenocarcinoma)	6.25–50 μg/mL (24 h & 48 h)	Exhibited antiproliferative activity	Antioxidant effects	Shen et al., 2019 [111]
Nuciferine, 7-hydroxydihydro-nuciferine, caaverine, liridodenine & anonaine	DU-145 (human prostate cancer)	IC_50_ = 80.8–218.4 μM (24 h)	Induced cytotoxicity	Not reported	Liu et al., 2014 [105]
Neferine	PC3; LNCaP (human prostate cancer); CD 44+ PC3 CSC (cancer stem cells)	3.12–100 µM (24–72 h)	Reduced cell proliferation and inhibited migration	↑Apoptosis; ↑G1 phase cell cycle arrest; ↓Bcl-2; ↓CDK4; ↑caspase-3; ↑cleaved PARP; ↑p21; ↑p27; ↑p53; ↓MMP-9; ↓Slug; ↓Snail; ↓SOD1; ↓CAT; ↓GPx	Erdogan & Turkekul, 2020 [132]
Neferine	DU145 and LNCaP (human prostate cancer)	5–20 µM (18 h)	Reduced cell viability	↑Apoptosis; ↑autophagy; ↓p62; ↑LC3B-II; ↑autophagosome formation; ↑p-JNK	Nazim et al., 2020 [133]
Neferine, liensinine, isoliensinine	LNCaP; PC3; DU-145 (human prostate cancer)	1–100 µM (24 and 48 h)	Induced cytotoxicity and reduced migration in PC3 and DU145 cells	↑Apoptosis; ↑autophagy; ↑Bax; ↓Bcl-2; ↑cleaved-caspase-9; ↑cleaved-PARP; ↓PARP; ↑LC3B-II; ↑AR; ↑PSA; ↑5-α reductase	Liu et al., 2021 [134]
*Renal cancer*					
Neferine	Caki-1 (human renal cancer)	5–25 µM (24 h)	Inhibited cell proliferation	↑Apoptosis; ↓Bcl-2; ↑Bax; ↓XIAP; ↓sub-G1 cell population; ↓NF-κB-dependent luciferase activity; ↓p65	Kim et al., 2019 [135]
*Sarcoma*					
Neferine	U2OS and Saos-2 (human osteosarcoma)	1–20 µM (24–72 h)	Inhibited cell proliferation	↑G1 arrest; ↓cyclin E; ↑p21; ↑p38 MAPK; ↑JNK	Zhang et al., 2012 [136]
*Skin cancer*					
Aqueous rhizome extract	A431 (epidermoid cancer)	1–1000 µg/mL (24 h)	Inhibited proliferation and migration	↓MMP-2; ↓MMP-9	Karki et al., 2008 [78]
Methanolic extracts from flower bud and leaves	B16 melanoma 4A5 cells (murine melanoma)	3–30 μM (72 h)	Inhibited melanogenesis	↓Tyrosinase; ↓TRP-1; ↓TRP-2	Nakamura et al., 2013 [66]
Procyanidin extract from seedpod	B16 (murine melanoma)	25–100 μg/mL (1–5 days)	Displayed cytotoxicity and inhibited cell proliferation	↑Apoptosis; ↑S cell cycle arrest; ↑calcium	Duan et al., 2010 [137]
Leaf extract; gallic acid	B16F1 (murine melanoma)	NLE: 0.1–0.5 mg/mL GA: 60–100 µM (24–72 h)	Reduced melanogenesis	↓Tyrosinase; ↓MITF; TRP-1; ↓p-PKA; ↓p-CREB; ↓melanin	Lai et al., 2020 [138]
7-Hydroxydehydronuciferine	A375.S2 (human melanoma)	10–100 µM (24 h)	Inhibited cell proliferation and showed cytotoxicity	↑Apoptosis	Liu et al., 2014 [105]
7-Hydroxydehydronuciferine	A375.S2, A375 and A2058 cells (human melanoma)	10–100 μM (24 h)	Induced cytotoxicity and reduced migration	↑Apoptosis; ↑autophagy; G2/M arrest; ↑ATG-5; ↑ATG-12; ↑ATG-16; ↑AVO	Wu et al., 2015 [139]

**Table 3 cancers-14-00529-t003:** Potential anticancer effects and mechanisms of action of *N. nucifera*-derived constituents based on in vivo studies.

Materials Tested	Animal Tumor Models	Anticancer Effects	Mechanisms	Dose (Route)	Duration	References
*Breast cancer*						
Flavonoid-rich leaf extract	BALB/c athymic nude mice injected with MCF-7 cells	Reduced tumor volume and weight	↓HER2; p-HER2; ↓Fas	0.5 & 1% (diet)	28 days	Yang et al., 2011 [79]
Aqueous leaf extract	MDA-MB-231 cells injected in female C57BL/6 nude mice	Inhibited tumor growth	Not reported	0.5–2 % (s.c.)	14 days	Chang et al., 2016 [80]
Liensinine + doxorubicin	Female nude mice injected with MDA-MB-231 cells	Reduced tumor growth	↑Apoptosis; ↑cleaved caspase-3; ↓autophagy/mitophagy; ↑auto-phagosome /mitophagosome; ↑colocalization of DNM1L and TOMM20	60 mg/kg (i.p.); 2 mg/kg (i.p.)	30 days	Zhou et al., 2015 [90]
*Colon cancer*						
Nuciferine	CT29 cells subcutaneously implanted in nude mice	Reduced tumor weight	Not reported	9.5 mg/kg (i.p.)	3 times a week for 3 weeks	Qi et al., 2016 [96]
Liensinine	HT29 cells injected in female BALB/c nude mice	Suppressed colorectal tumorigenesis, reduced tumor size	↓Ki-67	30 mg/kg (oral)	Every other day for 15 days	Wang et al., 2018 [97]
*Eye cancer*						
Neferine	WERI-Rb-1 cells injected in female athymic nude mice	Reduced tumor volume and weight	↓Ki-67; ↓VEGF; ↓SOD; ↑MDA	0.5–2 mg/kg (i.p)	Every 3 days for 30 days	Wang et al., 2020 [100]
*Gallbladder cancer*						
Liensinine	NOZ cells injected in BALB/c nude mice	Reduced tumor volume and weight	↓Ki-67	2 mg/kg (i.p)	Every 2 days	Shen et al., 2019 [101]
*Gastric cancer*						
Liensinine from seeds	SGC7901 cells injected in BALB/c homozygous (nu/nu) nude mice	Reduced tumor size	↓Ki-67	10 µM (i.p.)	Every 2 days for a month	Yang et al., 2019 [106]
*Head and neck cancers*						
Neferine	CAL27 cells injected in male BALB/c nude mice	Reduced tumor volume	↑Apoptosis; ↑autophagy, ↑cleaved caspase-3, ↑cleaved PARP1, ↑LC3; ↑p62	10 mg/kg (i.p)	Not reported	Zhu et al., 2021 [107]
*Liver cancer*						
Water-soluble polysaccharides from seeds	H22 cells injected in female Kunming mice	Reduced tumor weight	↑TNF-ɑ; ↑IL-2; ↑SOD; ↓MDA	50–200 mg/kg (oral)	14 days	Zheng et al., 2016 [102]
Leaf extract	DEN fed male Sprague-Dawley rats	Reduced tumor size	↓AST; ↓ALT; ↓albumin; ↓total triglyceride; ↓total cholesterol; ↓lipid peroxidation; ↑GSH; ↑GSHPx; ↑SOD; ↑CAT; ↑GST; ↓Rac1; ↓PKCɑ; ↓TNF-ɑ; ↓IL-6	0.5–2.0% (p.o.)	12 weeks	Horng et al., 2017 [119]
Leaf extract	2-AAF-induced male Wistar rats	Inhibited hepatic fibrosis and hepatocarcinogenesis	↓Triglycerides; ↓total cholesterol; ↓AFP; ↓IL-6; ↓TNF-ɑ; ↓AST; ↓ALT; ↓γGT; ↓GST-Pi; ↓lipid peroxidation; ↓8-OHdG; ↑Nrf2; ↑CAT; ↑GPx; ↑SOD-1	0.5–2% in the diet (p.o.)	6 months	Yang et al., 2019 [120]
Neferine+oxaliplatin	HepG2 and Bel-7402 cells injected in male BALB/c mice	Increased tumor volume reducing the effect of oxaliplatin	↑E-cadherin; ↓Vimentin; ↓Ki-67;	20 mg/kg/d (i.p.)	3 weeks	Deng et al., 2017 [116]
Isoliensinine	Huh-7 cells injected in male athymic nude mice and H22 cells injected in Kunming mice	Reduced tumor volume	↑caspase-3; ↓Bcl-2; ↓Bcl-xL; ↓MMP-9; ↓p65 phosphorylation	3 and 10 mg/kg/d (i.p. and gavage)	10 days; 3 weeks	Shu et al., 2015 [117]
Isoliensinine	Huh-7 cells transfectants injected in male athymic nude mice	Reduced tumor growth	↑Caspase-3 activity	10 mg/kg/d (gavage)	20 days	Shu et al., 2016 [118]
*Lung cancer*						
Leaf extract and leaf polyphenol extract	4T-1 metastatic tumor in the lung of BALB/c mice	Reduced metastasis and tumor weight	↓PKCɑ activation	0.25, 1% (p.o.)	19 days	Wu et al., 2017 [81]
Nuciferine	A549 cells injected in BALB/c mice	Reduced tumor size and weight	↑Apoptosis; ↓Bcl-2; ↑Bax; ↓Wnt/*β*-catenin; ↑Axin	50 mg/kg (i.p.)	3 times a week for 20 days	Liu et al., 2015 [126]
Neferine	DEN-induced lung carcinogenesis in albino male Wistar rats	Suppressed tumor growth	↓ROS; ↓lipid peroxidation; ↓protein carbonyl; ↑GSH; ↑SOD; ↑GPx; ↑GST; ↑CAT; ↓glycoprotein components; ↑ATPase; ↑p53; ↑Bax; ↑caspase-9; ↑caspase-3; ↓Bcl-2; ↓COX-2; ↓NF-κB; ↓CYP2E1; ↓VEGF; ↓PI3K; ↓Akt; ↓mTOR	10–20 mg/kg (oral)	20 alternate days	Sivalingam et al., 2019 [127]
*Neural cancer*						
Nuciferine	SY5Y cells subcutaneously implanted in nude mice	Reduced tumor weight	Not reported	9.5 mg/kg (i.p.)	3 times a week for 3 weeks	Qi et al., 2016 [96]
Nuciferine	U251 cells subcutaneously inoculated in BALB/c nude mice	Suppressed tumor weight and size	↓Ki-67; ↓CDC2; ↓Bcl-2; ↓HIF1A; ↓N-cadherin; ↓VEGFA	15 mg/kg (i.p.)	Once a day for 2 weeks	Li et al., 2019 [130]
*Skin cancer*						
Procyanidin extract from seedpod	B16 cells inoculated into syngeneic C57BL/6 J mice	Suppressed tumor volume and weight	↓lipid peroxidation levels; ↑SOD; ↑CAT; ↑GSPx; ↑spleen and thymus index	60–120 mg/kg (i.g.)	Every 2–3 days for 15 days	Duan et al., 2010 [137]
Leaf extract	UV-radiation exposed female guinea pigs	Reversed UVB-induced epidermal hyperplasia and hyperpigmentation	↓MITF; ↓tyrosinase; ↓TRP-1; ↓PKA; ↓ERK; ↓melanin	1–2% (topical)	2 weeks	Lai et al., 2020 [138]
7-Hydroxy-dehydronuciferine	A375.S2 cells injected in BALB/c nu/nu female mice	Reduced tumor volume	Not reported	20 mg/kg (i.p.)	Every 7 days for 28 days	Wu et al., 2015 [139]

#### 3.2.13. Nasopharyngeal Cancer

Zhao et al. [128] treated CNE-1 human nasopharyngeal carcinoma cells with alkaloids extracted from the Ba lotus (a new variety of *N. nucifera*) seeds for 24 h. The alkaloids reduced cell proliferation by increasing apoptosis and protein levels of apoptosis-associated factors, such as caspase-3, caspase-8, caspase-9, Bax, Fas, and FasL, and decreasing protein levels of antiapoptotic factors, such as Bcl-2 and Bcl-xL. The alkaloids also reduced the expression of NF-κB and increased the inhibitor of κBα (IκBα) levels to increase cell death and decrease cell growth.

#### 3.2.14. Neural Cancer

There are at least three reports on the in vitro evaluation of *N. nucifera* pure compounds against malignant nervous system tumors. Pham et al. [129] observed that neferine disrupted the growth of IMR32 human neuroblastoma cells by the induction of G2/M phase arrest, apoptosis and autophagy, cleavage of caspase-3 and PARP, accumulation of LC3-II, overexpression of Beclin-1, and reduction in phosphorylated FAK (p-FAK) and ribosomal S6 kinase 1 (p-S6K1). Additional results showed that neferine inhibited the migration of IMR32 cells. All these results were comparable to those of the standard anticancer drug temozolomide. Qi et al. [96] treated SY5Y human neuroblastoma cells with nuciferine and found that the alkaloid reduced cell viability and inhibited cell invasion. The mechanistic study revealed decreased expressions of PI3K, IL-1β, and p-Akt. Another research group, Li et al. [130], also investigated the anticancer properties of nuciferine in U87MG and U251 human glioblastoma cells. Nuciferine inhibited cell proliferation by increasing G2 cell cycle arrest and induced apoptosis by reducing the expression of Bcl-2 and increasing the expression of Bax. The alkaloid also reduced expression of hypoxia-inducible factor 1α (HIF-1α) and VEGFA in nuciferine-treated cells. Epithelial-to-mesenchymal transition was inhibited as nuciferine downregulated the expression of vimentin, N-cadherin, and snail family zinc finger 2 (Slug), yet upregulated E-cadherin and cyclin B1. Other proteins inhibited by nuciferine included p-Akt, phosphorylated signal transducer and activator of transcription 3 (p-STAT3), sex-determining region Y-box 2 (SOX2), and cell division control 2 (CDC2). 

Qi et al. [96] also used an in vivo approach to test the anticancer potential of nuciferine against neuroblastoma. Nude mice were subcutaneously implanted with SY5Y cells and injected with nuciferine 3 times a week for 3 weeks. The alkaloid resulted in reduced tumor weights. No mechanistic study was performed. In the in vivo study conducted by Li et al. [87], U251 cells were subcutaneously inoculated into BALB/c nude mice, and the mice with tumors were intraperitoneally injected with nuciferine. Nuciferine suppressed tumor weight and size by downregulating the expression of various proteins, such as Ki-67, Bcl-2, CDC2, HIF-1α, and N-cadherin. Interestingly, ultrasonography evaluation detected reduced tumor size, angiogenesis and hardness index in the nuciferine-treated group compared to control. Magnetic resonance imaging revealed similar results with tumor growth.

#### 3.2.15. Ovarian Cancer 

The potential anticancer effects of neferine have been observed in various ovarian cancer cell lines. Xu et al. [131] studied growth-inhibitory effects of neferine on the A2780 ovarian cancer cell line and non-malignant FTE187 immortalized fallopian epithelial cell line. The results showed that neferine exhibited inhibition of the proliferation of A2780 cells, whereas less cytotoxicity was observed for FTE187 cells. Neferine substantially reduced the colony-forming abilities of various ovarian cancer cell lines, such as A2780, SKOV3, and HO8910. In A2780 cells, neferine inhibited cell cycle progression at the G1 phase via the upregulation of G1/S cell cycle proteins p21 and p27 and decreased expression of cyclin D1 in A2780 cells. Neferine also showed apoptotic potential through autophagosome formation and increased expression of GFP-LC3. The expressions of autophagy pathway biomarkers, LC-III and Atg7, were also increased in neferine-treated A2780 cells. Additionally, mTOR-dependent autophagy was observed through decreased phosphorylated levels of p70S6K and 4EBP1 in neferine-exposed A2780 cells. Finally, neferine-induced autophagy through the activation of p38 MAPK and JNK pathways was observed in A2780 and HO8910 cells.

#### 3.2.16. Prostate Cancer

Several studies evaluated the anticancer properties of *N. nucifera* against prostate cancer. Shen et al. [111] explored the anticancer effects of the phenolic extract from the seedpods of *N. nucifera*. Phytochemicals isolated from the extract were catechin, hyperoside, kempherol, and quercetin. The phenolic seed extract induced antiproliferative activity in LNCaP human prostate adenocarcinoma cells. The seed extracts also exhibited antioxidant properties by eliciting free radical-reducing power, metal chelating capacity as well as inhibiting linoleic acid peroxidation. 

In addition to extracts, the anticancer properties of several pure compounds abundantly found in *N. nucifera* were also studied. Liu et al. [105] evaluated the effects of various aporphine alkaloids extracted and purified from the leaves of *N. nucifera* against DU-145 human prostate cancer cells. The most active compound, 7-hydroxydehydronuciferine, showed significant cytotoxicity against DU-145 cells, but the underlying mechanism has not been reported. In another study, it was found that neferine inhibited the viability and reduced the proliferation of PC3 and LNCaP human prostate cancers cells and CD44+ cancer stem cells (CSC) isolated from PC3 cells by inducing apoptosis and cell cycle arrest at the G1 phase. These effects were due to the upregulation of p21, p27, p53, poly adenosine diphosphate-ribose polymerase (clePARP) and the downregulation of CDK4 and Bcl-2. In addition, neferine induced oxidative stress by significantly downregulating the mRNA expression of CAT, SOD1 and GPx. Neferine inhibited CSC migration by downregulating the expression of MMP-9 along with the transcription factors Slug and Snail. Neferine was also shown to modulate major signaling pathways by increasing phosphorylation of p38, JNK and MAPKs both in PC3 and CD44+ CSCs [132]. Nazim et al. [133] treated DU-145 and LNCap cells with various concentrations of neferine to determine its anticancer effects. The study revealed that neferine reduced cell viability significantly when combined with tumor necrosis factor-related apoptosis-inducing ligand (TRAIL). The mechanisms behind the synergistic effects of TRAIL and neferine are increased apoptosis and autophagy via increased autophagosome formation and increased levels of LC3B-II, and decreased levels of p62. Neferine was also shown to activate the JNK pathway and increase p-JNK protein expression. A recent study by Liu et al. [134] evaluated the anticancer effects of neferine, liensinine, and isoliensinine on LNCaP, PC3, and DU-145 human prostate cancer cells. These compounds induced cytotoxicity in all 3 cell lines and reduced migration, particularly in PC-3 and DU145 cells. In LNCaP cells, all 3 compounds triggered apoptosis and autophagy, downregulated the protein expression of androgen receptor (AR), prostate-specific androgen (PSA), and type II 5-α-reductase and inactivated the PI3K/Akt signaling pathway.

#### 3.2.17. Renal Cancer 

There is at least one study that evaluated the anticancer effects of *N. nucifera* in renal cancer. An investigation led by Kim et al. [135] explored the anticancer effects of neferine on Caki-1 renal cancer cells. Neferine was found to exhibit a growth-inhibitory effect in a concentration-dependent manner and caused apoptosis through the downregulation of Bcl-2 and X-linked inhibitor of apoptosis (XIAP). Further investigation revealed that the downregulation of Bcl-2 by neferine was mediated by the inhibition of the NF-κB pathway. In other renal cancer cell lines, such as ACHN and A498, neferine was also found to downregulate Bcl-2 expression, induce the cleavage of procaspase-3 and inhibit p65 expression. 

#### 3.2.18. Sarcoma 

There are relatively few studies that examined the effect of neferine in sarcoma. The antiproliferative effects of neferine on U2OS and Saos-2 human osteosarcoma cell lines were demonstrated in a study by Zhang et al. [136]. This research group observed that neferine significantly reduced proliferation in the U2OS and Saos-2 cell lines with increasing concentrations of nefereine; however, normal osteoblasts, HCO, were less sensitive. The results suggested that neferine caused osteosarcoma cells to arrest in the G1 phase of the cell cycle through the increase in p21 protein level, which was independent of p53 or retinoblastoma-associated protein activation. The study went on to find that neferine treatment activated p38 MAPK and JNK through phosphorylation, leading to p21 accumulation. 

#### 3.2.19. Skin Cancer 

There have been several studies evaluating the anticancer effects of *N. nucifera* on melanoma cells in vitro. Karki et al. [78] explored the anticancer effects of an aqueous rhizome extract on A431 epidermoid cancer cells. The extract was shown to inhibit cancer cell proliferation and migration by decreasing the levels of MMP-2 and MMP-9 in a concentration-dependent fashion. In another study, B16 melanoma 4A5 murine cells were treated with methanolic extracts from the flower and leaves of *N. nucifera*. These extracts inhibited cell viability by reducing the expression of tyrosinase, tyrosine-related protein-1 (TRP-1), and TRP-2. Based on phytochemical analysis, the methanolic extract yielded various constituents, such as nuciferine, N-methylasimilobine, and 2-hydroxy-1-methoxy-6a, 7-dehydroaporphine [66]. Another study investigated the anticancer effect of procyanidins extracted from the seedpod of *N. nucifera*. In this in vitro study, Duan et al. [137] treated B16 murine melanoma cells with the procyanidins, and the results showed cytotoxicity and inhibited cell proliferation via increased apoptotic activity, S phase cycle arrest, and increased intracellular calcium levels. In a recent study, NLE as well as its major component, gallic acid, induced cytotoxicity in B16F1 murine melanoma cells. Mechanistic studies using an immunoblotting assay revealed that both treatments reduced the expressions of tyrosinase, microphthalmia-associated transcription factor (MITF), TRP-1, phosphorylated protein kinase A (p-PKA), and phosphorylated cAMP response element-binding protein (p-CREB) (Lai et al., 2020). In another study, Liu et al. [138] demonstrated that 7-hydroxydehydronuciferine, a compound isolated from a methanolic extract of the leaves of *N. nucifera* Gaertn. cv. Rosa-plena, inhibited cell proliferation and showed cytotoxicity by increasing apoptosis in A375.S2 melanoma cells. Wu et al. [139] treated human melanoma cells (A375.S2, A372, and A2058) with various concentrations of 7-hydroxydehydronuciferine, isolated from the leaves of *N. nucifera*, and observed its anticancer effects. It was found that the compound induced cytotoxicity, registered G2/M phase cell cycle arrest, and reduced cellular migration in parallel with increased apoptosis, autophagy, and autophagy-related proteins (ATG-5, ATG-12, and ATG 16) as well as altered cellular acidic vesicular organelles (AVO).

In addition to their in vitro study, Duan et al. [137] also evaluated the anticancer effect of *N. nucifera* seedpod-derived procyanidins using an in vivo skin cancer model. This was performed by subcutaneously inoculating syngeneic C57BL/6 J mice with B16 melanoma cells and treating the animals with the procyanidin extract every 2–3 days for 15 days. The results showed suppressed tumor growth (reduced volume and weight) through the reduction of hepatic lipid peroxidation levels and the promotion of superoxide dismutase SOD, CAT, and GPx activity. The mice treated with the procyanidin-rich extract also displayed an increased spleen and thymus index, indicating immunomodulatory activities. Lai et al. [138] also conducted an in vivo study using ultra-violet B (UVB)-radiation exposed female guinea pigs and topically treated them with 1–2% NLE. The test agent was found to reverse UVB-induced epidermal hyperplasia and decrease melanin content, as evidenced by hematoxylin and eosin staining and Fontana-Masson staining, respectively. Western blot analysis showed that NLE downregulated the levels of MITF, tyrosinase, and TRP-1, and reduced the activity of PKA and ERK signaling. In addition to the in vitro study mentioned before, Wu et al. [139] studied the effects of 7-hydroxydehydronuciferine in vivo using BALB/c nu/nu female mice with xenotransplanted A375.S2 tumors. Intraperitoneal (i.p.) injections of the alkaloid (20 mg/kg) every 7 days for 28 days reduced tumor volume. The mechanisms behind these antitumor effects have not been reported.

## 4. Bioavailability and Pharmacokinetics of *N. nucifera* Constituents

Various pure compounds of *N. nucifera* have different bioavailability and pharmacokinetic characteristics. The pharmacokinetic profile of neferine, an alkaloid with significant antineoplastic activity as presented in this review, was studied by Zhao et al. [140] using canine models. Following oral administration, it was found that the plasma concentration of neferine peaked twice, first at 0.333 h and again at 0.667 h after intake. The absolute bioavailability of oral neferine was 65.36%. Another study [141] implemented rat models and also observed two absorption peaks of neferine after oral administration. The plasma concentration peaks were at 10 min and 1 h after intake. The greatest level of distribution of neferine in the liver was recorded when the rats were given 10 and 20 mg/kg of the alkaloid. However, when the rats were administered 50 mg/kg of neferine, its distribution was the greatest in the kidneys and lungs. In addition to these findings, the investigators also observed that neferine was metabolized by cytochrome P450 (CYP) 2D6 to yield metabolites, such as liensinine, isoliensinine, desmethyliensinine, and desmethyl-isoliensinine. A separate study evaluated the metabolism of isoliensinine identified using reversed-phase high-performance liquid chromatography and electrospray ionization tandem mass spectrometry. Using canine models, Zhou et al. [142] identified three metabolites of isoliensinine, namely 2-*N*-desmethyl-isoliensinine, 2′-*N*-desmethylisoliensinine, and 2′-*N*-6-*O*-didesmethylisoliensinine. These findings suggest that isoliensinine is primarily metabolized by *N*-demethylation and *O*-demethylation in the liver. Zhao et al. [143] also explored pharmacokinetic profiles of neferine and found it had little effect on various CYP enzymes, namely CYP1A2, CYP2D6, and CYP3A4. Kadioglu et al. [86] investigated the pharmacokinetics of neferine using an in silico absorption, distribution, metabolism, and excretion study. The values of logS for solubility, logP for the partition coefficient, and logD for the distribution coefficient were 4.302, 4.432 and 3.503, respectively, showing that neferine possessed the required druggability properties, such as aqueous solubility and lipophilicity. 

Other alkaloids derived from *N. nucifera*, such as nuciferine and *N*-nuciferine, were also investigated to determine their pharmacokinetic properties. Ye et al. [47] used male rat models and found that nuciferine and N-nuciferine were rapidly absorbed in the blood and reached a maximum plasma concentration of 1.71 and 0.57 μg/mL at 0.9 and 1.65 h, respectively. The oral bioavailability for nuciferine was 58.13%, while for *N*-nuciferine, it was 79.91%. The researchers also found that the alkaloids rapidly crossed the blood-brain barrier and achieved widespread distribution in the brain. Ye et al. [144] studied the inhibitory effects of nuciferine, *N*-nuciferine, and 2-hydroxy-1-methoxyaporphine on CYP enzymes. It was found that these three alkaloids inhibited the CYP2D6-catalyzed *O*-demethylation form of metabolism. This needs to be considered when administering such *N. nucifera* compounds to patients taking anti-arrhythmia drugs (such as amiodarone and metoprolol) that are specific substrates of CYP2D6. Zou et al. [145] isolated five alkaloids (nuciferine, *O*-nornuciferin, liriodenine, armepavine, and pronuciferine) from *N. nucifera* leaf extract that were then orally administered to rats. The plasma half-lives of nunciferine, O-nornuciferin, liriodenine, armepavine, and pronuciferine were 6.18 ± 3.10, 6.67 ± 2.88, 3.77 ± 1.15, 5.22 ± 5.09, and 4.44 ± 1.88, respectively. Additionally, after giving 2.4 g/kg of the *N. nucifera* leaf extract to rats, it was found that liriodenine and pronuciferine had the slowest absorption rate. Yan et al. [146] detected an alkaloid higenamine in the green embryo of the mature seeds of *N. nucifera*, also known as plumula nelumbinis, a commonly used traditional Chinese medicine. It is interesting that higenamine is included in the Word Anti-Doping Agency (WADA) 2017 Prohibited List and is therefore checked in urine samples of athletes undergoing drug tests. Yan et al. [145] had 14 human volunteers who ingested capsules containing 0.34 g of plumula nelumbinis, and 11 human volunteers ingested capsules with 5 mg higenamine for seven days. The participants who ingested the higenamine capsules showed a maximum concentration of 2000 ng/mL in their urine, rising above the WADA reporting limit of 10 ng/mL. Within 3–7 days of ingesting the plumula nelumbinis capsules, the participants also exceeded the WADA reporting limit with a maximum concentration of 500 ng/mL of higenamine in their urine. 

Overall, it has been found that multiple compounds of *N. nucifera* have high oral bioavailability, but caution needs to be taken due to their interactions with hepatic biotransformation enzymes. Further research using human models should be performed to determine the bioavailability and pharmacokinetics of *N. nucifera* compounds.

## 5. Toxicity Studies on *N. nucifera*

The lotus plant (*N. nucifera)* has been used in traditional medicine for many years and is consumed all around the world. Several studies have evaluated the toxicity and safety profile associated with *N. nucifera* and its constituents. There are various in vitro studies that used normal cell lines to determine the toxicity of *N. nucifera*. In one such study, a normal osteoblast cell line, HCO, was treated with neferine, and the results showed that the cell viability was not significantly affected [136]. Poornima et al. [123] treated BEAS-2B normal lung cells with neferine and found that there was no significant suppression of cell growth up to 20 µM. Xu et al. [130] used normal fallopian tube epithelial cells, FTE187, and observed that neferine did not significantly affect cell viability. In another study evaluating the toxicity of neferine, Yoon et al. [114] treated THLE-3, normal human liver cells, with 5–30 µM of the alkaloid for 24 h. The THLE-3 cells showed no cytotoxicity and change in viability even at 30 µM. Yang et al. [85] treated Hs578Bst mammary fibroblasts with neferine at concentrations up to 8 mg/mL and found that it did not affect cell growth and showed no toxicity. Liao and Lin [147] determined the safety profile of *N. nucifera* plumule polysaccharide in isolated mouse splenocytes. The results showed that the polysaccharide, even at a concentration of 125 µg/mL, did not significantly affect splenocyte viability. 

There are also multiple in vivo studies that have investigated the safety profile of *N. nucifera*. Kunanusorn et al. [148] explored the acute and subchronic oral toxicity of *N. nucifera* stamens extract using a rodent model. In this experiment, 7–8-week-old male and female Sprague-Dawley rats were administered 5000 mg/kg of the extract in the acute study and 50–200 mg/kg/day for 90 days in the subchronic study. Within 14 days of the acute study and days 90–118 in the subchronic study, no signs of toxicity or significant changes in weight were noted. However, after day 118, treatment of 200 mg/kg/day in the subchronic study showed an increased number of lymphocytes, decreased number of basophils, and decreased mean corpuscular volume in the male rats. Similarly, female rats showed a decrease in red blood cells and hematocrit after 90 days and an increase in the mean corpuscular hemoglobin concentration after 118 days. In addition, after 90 days of the subchronic treatment of 200 mg/kg/day, there was a decrease in creatinine and cholesterol in male rats and a decrease in albumin in female rats. Female rats also had an increased alkaline phosphatase after 118 days. Although there were no changes in the gross appearance of the internal organs, the relative kidney, liver, and heart organ weights were lower than the control on day 90 of 200 mg/kg/day. All these results support the conclusion that the lethal oral dose of the stamen extract is more than 5000 mg/kg, and the no-observed-adverse-effect level is greater than 200 mg/kg/day for 90 days. In another in vivo study, Rajput and Khan [149] administered 10–5000 mg/kg of *N. nucifera* seed pod extract in albino mice and found that all doses up to 5000 mg/kg were tolerated for 24–48 h. No death or behavioral abnormalities were observed. Another study also utilized mice which were intragastrically administered a water extract from *N. nucifera* leaf. The median lethal dose of the extract was observed to be greater than 5000 mg/kg [150]. One study used Beagle dogs to determine the safety profile of *N. nucifera* seed extract, which was administered for over 4 weeks, and found it to be safe at 2000 mg/kg/d. The research group also used rats and found the extract at 4000 mg/kg/d to be the safe dose over a 13-week administration period [151]. 

Kadioglu et al. [86] evaluated the toxicity of the pure *N. nucifera* compound neferine towards normal tissues using Stardrop and Derek Nexus software. The in silico safety evaluation showed a favorable toxicity profile. No mutagenicity toxicity class was predicted. However, it has been found that neferine could cause some skin sensitization. Overall, multiple studies show that *N. nucifera* is relatively safe to use at therapeutic doses. Nevertheless, additional in vivo studies and clinical evaluations are warranted to confirm the non-toxicity and safety of *N. nucifera* for human use.

## 6. Conclusions, Current Challenges/Limitations and Future Directions

All parts of *N. nucifera* have been used for food and medicinal purposes across Asia for over a thousand years. More recently, *N. nucifera* constituents have been studied for their chemoprotective and antineoplastic potential. In this systematic review, we have found extensive evidence of mechanism-based cancer preventive and therapeutic effects of bioactive constituents from different parts of *N. nucifera*, as well as the bioavailability, pharmacokinetics, and toxicities of selected phytochemicals. 

We analyzed a multitude of antineoplastic effects of *N. nucifera* extracts, fractions and pure compounds on various cancer types. Specific phytochemicals, including neferine, nuciferine, liensinine, 7-hydroxydehydronuciferine, cadaverine, liriodenine, anonaine, and gallic acid, have been investigated for their anticancer effects. *N. nucifera*-derived products and phytochemicals exhibit antineoplastic effects via antioxidant, anti-inflammatory, antiproliferative, cell death-inducing, cell cycle-regulatory, anti-invasive, and antiangiogenic pathways (Figure 5). *N. nucifera* phytochemicals prevented oxidative stress by altering GSH, iNOS, and CAT and modulated various proteins involved in inflammatory processes (NF-κB, COX-2, and TLR-4), proliferation (cyclins, cyclin-dependent kinases, and MAPK), apoptosis (Bax, Bad, caspase-3, caspase-6, caspase-7, caspase-8, caspase-9 and Bcl-2), invasion (MMP-2, N-cadherin, snail, and slug), and angiogenesis (VEGF and CTGF). Regarding in vitro studies, apoptosis was induced by various *N. nucifera* compounds across the majority of cancer subtypes. Various anticancer mechanisms involved activated caspase-3, caspase-6, caspase-7, caspase-8, and caspase-9, increased Bax and Bad, and decreased Bcl-2, indicating that *N. nucifera* phytochemicals induce apoptosis through extrinsic and intrinsic pathways. In vivo studies included various mechanisms without overlap between cancer subtypes.

The most-studied *N. nucifera* phytochemical has been neferine, a bis-benzylisoqunoline alkaloid, isolated from seed pod embryos. The majority of the chemotherapeutic literature on *N. nucifera* is limited to in vitro studies, as there are 72 in vitro studies and only 23 in vivo studies. The quality of each animal study was calculated using the SYRCLE’s RoB tool [76]. The 17 in vivo studies had an average of 32.2% for the quality control analysis, since many protocol details were not stated. The cancer types researched include breast, colon, gastric, chronic myelocytic leukemia, laryngeal, liver, lung, nasopharyngeal, ovarian, prostate, renal and skin cancers. The most thoroughly studied cancers included gastrointestinal, breast, and lung cancers in vitro as well as in vivo studies of gastrointestinal cancers.

We have identified various limitations in the current research, including the limited bioavailability and sparse toxicology research. There are limited studies on the bioavailability of *N. nucifera*; however, some literature shows the relatively high bioavailability and distribution of *N. nucifera* constituents nuciferine, *N*-nuciferine, and neferine. *N. nucifera* compounds were also shown to interact with various hepatic CYP enzymes, which could interfere with other pharmaceuticals or chemotherapeutic drugs if used in combination with *N. nucifera*. Limited toxicity studies conclude that *N. nucifera* plumule polysaccharides and neferine are nontoxic and safe for use. Due to its limited toxicity, *N. nucifera* can pose as an alternative or in conjunction with traditional chemotherapy, which has numerous undesirable side effects. 

Our group has identified many avenues of further research. Firstly, more in vivo studies should be conducted as at present, the majority of the research is limited to in vitro studies. The in vivo studies should include additional cancer subtypes and validate in vitro mechanisms of action. Currently, no clinical research has been conducted to explore the anticancer potential of *N. nucifera* constituents. Hence, randomized controlled trials are necessary to translate preclinical results into clinical practice. Another avenue of further research includes studies on synergistic effects with other anticancer drugs, as the current literature mostly examines *N. nucifera* phytochemicals individually. Along with the existing research on anticancer mechanisms of *N. nucifera*, additional research is necessary to ascertain the molecular targets of the parent compound and their metabolites in various organs. Supplementary research is needed to optimize delivery systems for *N. nucifera*’s active phytocompounds, which would increase their bioavailability and anticancer effects via efficient targeting. The quantity of *N. nucifera*-derived food items that needs to be consumed for prevention of various cancers is also yet to be determined and needs further research. Although several studies include the synergistic effects of *N. nucifera* phytochemicals and traditional chemotherapeutic drugs, such as cisplatin and oxaliplatin, additional research should be conducted to include different types of cancer as well as expansion to clinical studies. Taking into consideration the in-depth analysis of current research as presented in this review, *N. nucifera*-derived bioactive phytochemicals possess significant potential for human cancer prevention and anticancer therapy.

## Figures and Tables

**Figure 1 cancers-14-00529-f001:**
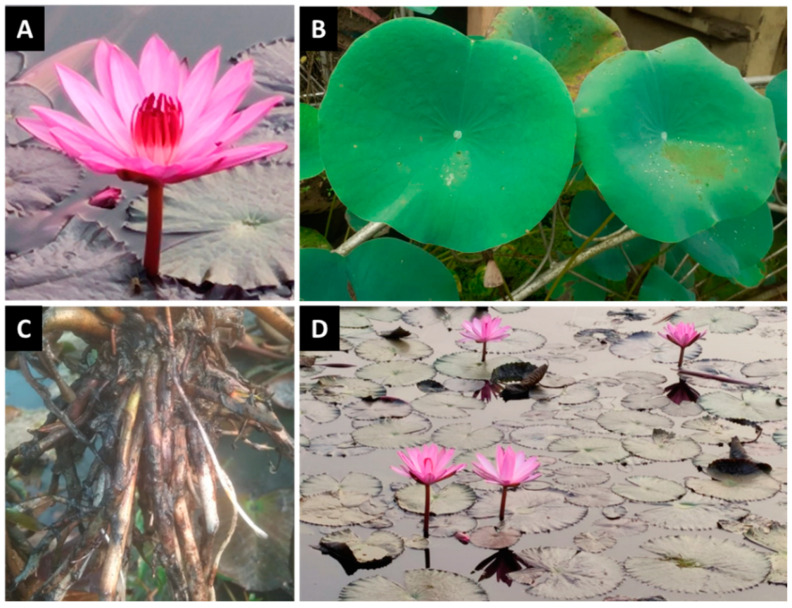
Photographs of lotus (*N. nucifera*). (**A**) Flower; (**B**) leaves; (**C**) rhizomes and (**D**) natural habitat.

**Figure 2 cancers-14-00529-f002:**
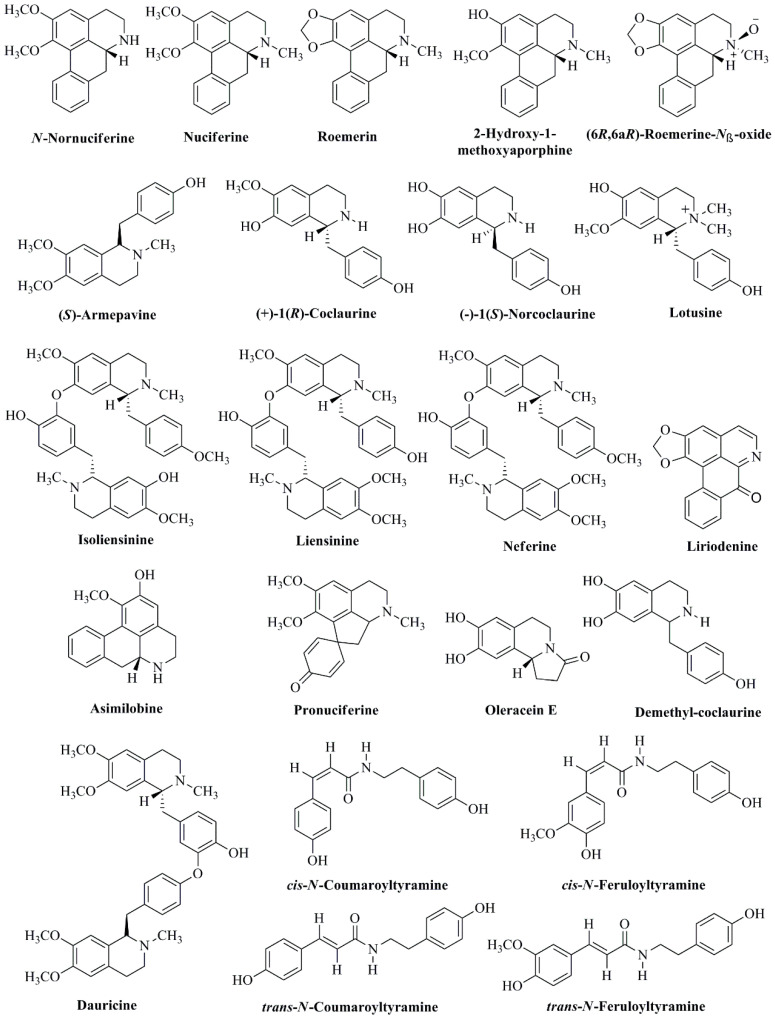
Chemical structures of major bioactive alkaloids isolated from different parts of *N. nucifera*.

**Figure 3 cancers-14-00529-f003:**
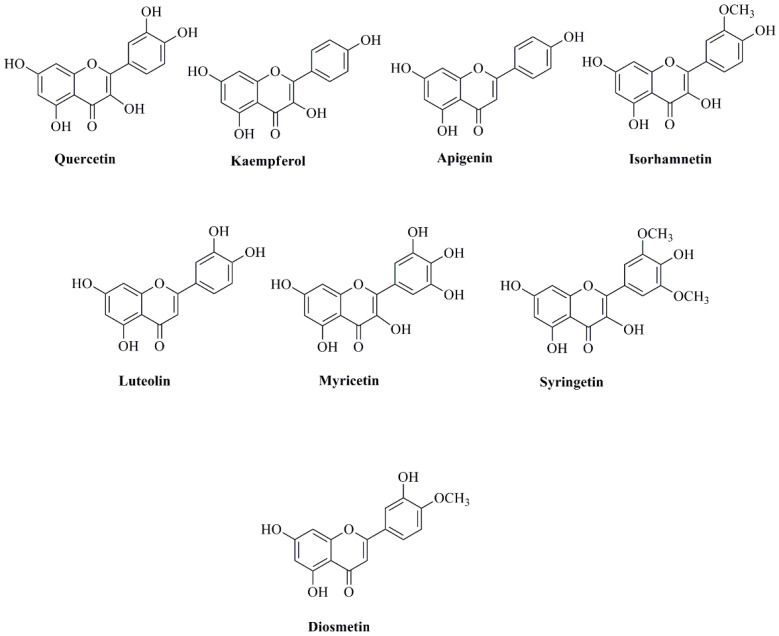
Chemical structures of major bioactive flavonoids isolated from different parts of *N. nucifera*.

**Figure 4 cancers-14-00529-f004:**
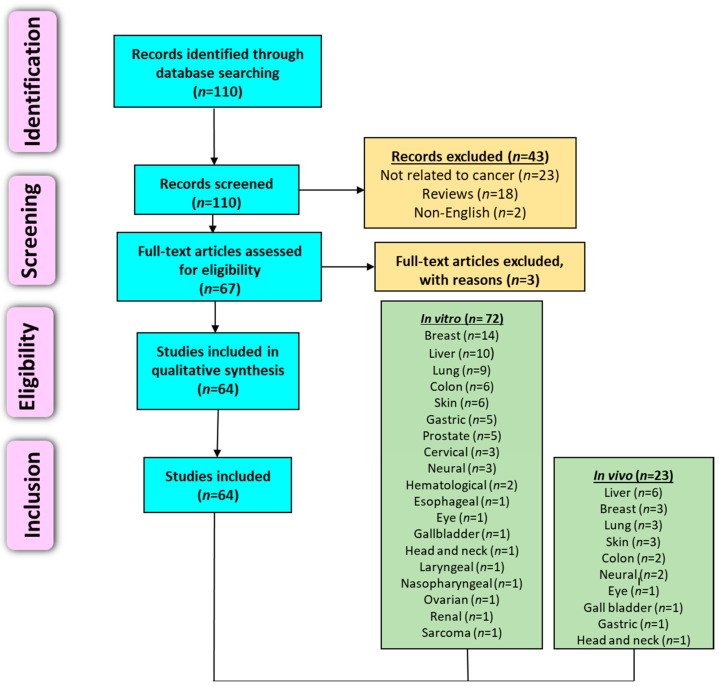
A PRISMA flowchart illustrating the literature search and study selection process relevant to the anticancer potential of *N. nucifera*. The total number of in vitro and in vivo studies 95 is greater than the number of studies included in this work (64) because many publications contained results based on more than one organ-specific cancer or study type (in vitro and in vivo).

**Figure 5 cancers-14-00529-f005:**
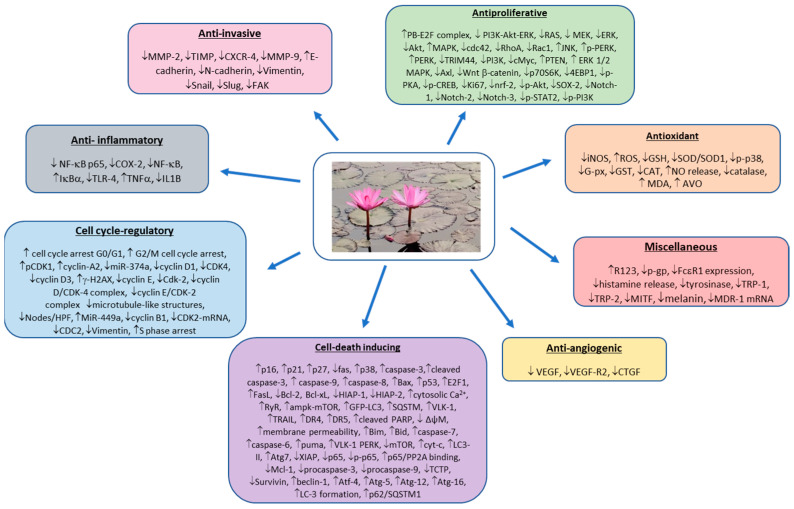
Overview of anticancer mechanisms, molecular targets, and signaling pathways of *N. nucifera* phytochemicals based on in vitro and studies.

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
