# Peer review of "Lotus (Nelumbo nucifera Gaertn.) and Its Bioactive Phytocompounds: A Tribute to Cancer Prevention and Intervention"

_cancers, 2022, doi:10.3390/cancers14030529_

Round 1

Reviewer 1 Report

Bishayee et al performed a systematic review and a critical analysis of various studies to provide an up-to-date assessment of cancer preventive and anticancer therapeutic attributes of N. nucifera and its bioactive phyto-components. Overall, the review is informative. The tables are particularly useful, as they provide a comprehensive listing of phytochemicals and their mechanisms of action in various types of cancers. I have a few minor suggestions:

Line 334: the structure of the paragraph should be 3.2.1 , 3.2.2 etc.

Line 340 and 444: contain the only reference to Table 2 and Table 3 respectively and are too general. I would add a sentence "Potential anticancer effects and mechanisms of action of N. nucifera-derived constituents based on in vivo studies have been shown in Table 2 and Table 3" and moved the tables at the end of the paragraph as a summary. 

Author Response

The authors of this manuscript express their sincere thanks to the reviewer for the critical assessment of this work. The authors have acted upon the recommendations of the reviewers which have resulted in a significant enhancement in the quality of this manuscript. All modifications incorporated in the manuscript are highlighted in red color font. A “point-by-point” response to each and every comment is outlined below.

General comments:

Bishayee et al performed a systematic review and a critical analysis of various studies to provide an up-to-date assessment of cancer preventive and anticancer therapeutic attributes of N. nucifera and its bioactive phyto-components. Overall, the review is informative. The tables are particularly useful, as they provide a comprehensive listing of phytochemicals and their mechanisms of action in various types of cancers. I have a few minor suggestions:

Response:

Thank you for your expertise, time, and effort for reviewing our manuscript. We are deeply encouraged by the generous comments regarding the quality of our work. We sincerely appreciate the valuable suggestions which we have found extremely valuable while revising our manuscript.

Specific comments:

Comment 1:

Line 334: the structure of the paragraph should be 3.2.1 , 3.2.2 etc.

Response:

We sincerely apologize for the inadvertent errors and corrected the subsection numbers (page 27, line 380; page 29, line 508; page 30, line 583; page 30, line 593; page 31, line 611; page 31, line 625; line 32, line 659; page 32, line 676; page 32, line 693; page 33, line 703; page 34, line 797; page 36, line 890; page 36, line 899; page 37, line 931, page 37, line 948; page 38, line 985; page 38, line 995; and page 38, line 1007).

Comment 2:

Line 340 and 444: contain the only reference to Table 2 and Table 3 respectively and are too general. I would add a sentence "Potential anticancer effects and mechanisms of action of N. nucifera-derived constituents based on in vivo studies have been shown in Table 2 and Table 3" and moved the tables at the end of the paragraph as a summary.

Response:

We agree with the excellent suggestion of the reviewer. We have added introductory text to cite Table 2 and Table 3 (page 15, lines 351-354). We have also moved these tables after the introductory paragraph as suggested.

Additionally,

The entire manuscript has been thoroughly checked and edited to minimize typographical errors as well as to ensure uniform style, organization, and quality.

Finally,

On behalf of my co-authors, I once again express my sincere thanks to the erudite Associate Editor and reviewers for the valuable suggestions and constructive input to improve the quality of our manuscript.

Reviewer 2 Report

This review article evaluated the potential of N. nucifera phytoconstituents in cancer prevention and therapy with reference to the potential mechanisms of action. There are some concerns in this manuscript as follows:

  1. The novel points in this review article should be clarified because there are previous studies that demonstrated the anticancer effects of N. nucifera phytoconstituents.
  2. Too many references (11 references) were put at the end of the first paragraph page 2 line 57. The number of these references should be reduced
  3. Page 3 Lines 126-133: The reference [36] was repeated in successive manner. I think addition of different references for each sentence in this part would be more appropriate.
  4. The meaning of the abbreviations should be clearly defined at their first mention.
  5. Too much data is put in tables 1 and 2. I think it may be better to divide these collective tables into small tables to be less confusing to the readers.
  6. I think that the conclusion should be summarized to focus on the possible clinical implications of the data obtained from the present review.
  7. The manuscript should be checked regarding the grammatical and typing errors.

Author Response

The authors of this manuscript express their sincere thanks to the reviewer for the critical assessment of this work. The authors have acted upon the recommendations of the reviewers which have resulted in a significant enhancement in the quality of this manuscript. All modifications incorporated in the manuscript are highlighted in red color font. A “point-by-point” response to each and every comment is outlined below.

General comments:

This review article evaluated the potential of N. nucifera phytoconstituents in cancer prevention and therapy with reference to the potential mechanisms of action. There are some concerns in this manuscript as follows:

Response:

Thank you for your expertise, time, and effort for reviewing our manuscript. We sincerely appreciate the valuable suggestions which we have found extremely valuable to improve the quality of our manuscript.

Specific comments:

Comment 1:

The novel points in this review article should be clarified because there are previous studies that demonstrated the anticancer effects of N. nucifera phytoconstituents.

Response:

We thank the reviewer for this thought-provoking comment. In the last paragraph of the introduction section, we have cited previous relevant reviews and discussed their limitations. We have also clearly presented the novelty and possible impact of our work (page 2, line 90 to page 3, line 105).

Comment 2:

Too many references (11 references) were put at the end of the first paragraph page 2 line 57. The number of these references should be reduced

Response:

While we understand the concern of the reviewer, the cited references are needed to support the statement regarding critical previous work on the effects of phytochemicals in modulating various cancer hallmarks and associated signaling pathways. We have carefully chosen only citations of authoritative articles published in high-impact journals.

Comment 3:

Page 3 Lines 126-133: The reference [36] was repeated in successive manner. I think addition of different references for each sentence in this part would be more appropriate.

Response:

We thank the reviewer for this close observation. We have condensed the description to limit multiple citations (page 3, line 128 to page 4, line 134).

Comment 4:

The meaning of the abbreviations should be clearly defined at their first mention.

Response:

We have followed the “rule of abbreviations” throughout the manuscript. We have used full forms with acronyms in the parenthesis for the first time. Next time, we just used the acronym. A separate and comprehensive list of all abbreviations and their explanations has been provided before the reference section following the journal style (pages 43-45).

Comment 5:

Too much data is put in tables 1 and 2. I think it may be better to divide these collective tables into small tables to be less confusing to the readers.

Response:

We appreciate the reviewer’s comment. Tables 1 and 2 provide a comprehensive listing of phytochemicals and their potential anticancer effects as well as mechanisms of action in various types of cancers, respectively. These tables are fundamental aspects of our manuscript and are important distinguishing features of our comprehensive review of current literature. Breaking down of these tables would compromise the purpose as well as the quality our manuscript. We sincerely believe our readers would appreciate having all relevant data in one place for qualitative and quantitative comparison.

Comment 6:

I think that the conclusion should be summarized to focus on the possible clinical implications of the data obtained from the present review.

Response:

This is an excellent point. In spite of the publication of numerous studies on in vitro and in vivo anticancer effects of N. nucifera and its bioactive constituents, till date no clinical studies have been published. Hence, the full potential of preclinical data for clinical application is yet to be realized. We have highlighted this limitation and proposed future research directions (page 43, lines 1221-1223). However, we have revised the concluding statement to focus on the possible clinical implications of impressive preclinical data presented in this manuscript (page 43, lines 1235-1237).

Comment 7:

The manuscript should be checked regarding the grammatical and typing errors.

Response:

We are sorry for any inadvertent errors. We have thoroughly checked our manuscript and edited it to eliminate grammatical and typographical errors.

Additionally,

The entire manuscript has been thoroughly checked and edited to minimize typographical errors as well as to ensure uniform style, organization, and quality.

Finally,

On behalf of my co-authors, I once again express my sincere thanks to the erudite Associate Editor and reviewers for the valuable suggestions and constructive input to improve the quality of our manuscript.